# Evaluation of methods to determine the surface mixing layer height of the atmospheric boundary layer in the central Arctic during polar night and transition to polar day in cloudless and cloudy conditions

Elisa F. Akansu[1], Sandro Dahlke[2], Holger Siebert[1], and Manfred Wendisch[3]

[1]Leibniz Institute for Tropospheric Research (TROPOS), Leipzig, Germany
[2]Alfred Wegener Institute (AWI), Helmholtz Centre for Polar and Marine Research, Potsdam, Germany
[3]Leipzig Institute for Meteorology (LIM), Leipzig University, Leipzig, Germany

**Correspondence:** Elisa F. Akansu (akansu@tropos.de)

**Abstract.** This study evaluates methods to derive the surface mixing layer (SML) height of the Arctic atmospheric boundary layer (ABL) using in situ measurements inside the Arctic ABL during winter and the transition period to spring. An instrumental payload carried by a tethered balloon was used for the measurements between December 2019 and May 2020 during the yearlong Multidisciplinary drifting Observatory for the Study of Arctic Climate (MOSAiC) expedition. Vertically highly resolved (cm-scale) in situ profile measurements of mean and turbulent parameters were obtained, reaching from the sea ice to several hundred meters height above ground.

Two typical conditions of the Arctic ABL over sea ice were identified: cloudless situations with a shallow surface-based inversion and cloudy conditions with an elevated inversion. Both conditions are associated with significantly different SML heights whose determination as accurately as possible is of great importance for many applications. We used the measured turbulence profile data to define a reference of the SML height. With this reference, a more precise critical bulk Richardson number of 0.12 was derived, which allows an extension of the SML height determination to regular radiosoundings. Furthermore, we have tested the applicability of the Monin-Obukhov similarity theory to derive SML heights based on measured turbulent surface fluxes. The application of the different approaches and their advantages and disadvantages are discussed.

## 1 Introduction

Currently, the Arctic climate is changing rapidly, driven by intertwined mechanisms and feedbacks, such as the lapse-rate feedback and surface-albedo feedback, leading to an increased near-surface air temperature and corresponding sea ice retreat (Serreze and Barry, 2011; Wendisch et al., 2023a). Atmospheric and cloud processes contribute significantly to these ongoing climate changes in the Arctic (Wendisch et al., 2019). The enhanced response of the Arctic climate system to global warming is referred to as Arctic amplification. There are still significant gaps in understanding this phenomenon, causing major uncertainties in projections of the future Arctic climate (Cohen et al., 2020). In particular, the processes determining the evolution of the Arctic atmospheric boundary layer (ABL) in cloudless and cloudy situations are not well represented by weather/climate models (Wendisch et al., 2019). The ABL is the atmospheric layer above the Earth's surface that is directly influenced by the

surface (Stull, 1988; Garratt, 1997). Especially during polar night, the vertical extent of the ABL plays an important role as stable stratification hampers the vertical exchange of energy and leads to a near-surface warming contributing to Arctic amplification mostly in winter (Graversen et al., 2008; Bintanja et al., 2011). Models often fail in reproducing shallow ABLs in the Arctic (Esau and Zilitinkevich, 2010; Lüpkes et al., 2010).

To advance our knowledge of the vertical structure of the ABL, tethered balloon-borne observations were performed during the yearlong Multidisciplinary Drifting Observatory for the Study of Arctic Climate (MOSAiC) expedition from October 2019 to September 2020 (Shupe et al., 2022). Profiling the lower atmosphere with an instrumental system carried by a tethered balloon provides high resolution (125 Hz) in situ data throughout the ABL, reaching from the sea ice surface up to several hundred meters height above ground.

The Arctic ABL is formed under unique conditions, such as the strong cooling of the sea ice surface due to the lack of solar radiation during winter, which favors the evolution of stable atmospheric layering. Furthermore, the ABL does not develop a residual layer due to the absence of a diurnal cycle for most of the year, and even during the polar day, convection typically plays a minor role (Persson et al., 2002; Tjernström and Graversen, 2009; Morrison et al., 2012; Brooks et al., 2017). Within this study, we refer to two typical states of the Arctic ABL observed over sea ice in winter and early spring: a cloudless ABL with a surface-based temperature inversion, and a cloudy ABL with a pronounced cloud top inversion (Tjernström and Graversen, 2009; Stramler et al., 2011; Morrison et al., 2012; Wendisch et al., 2023b). We have distinguished between those two states of the atmosphere, as many models have difficulties to reproduce the bimodal distribution of the terrestrial radiation (Solomon et al., 2023).

Under cloudless conditions, the atmosphere cools radiatively from the surface (strong negative net thermal-infrared irradiances), forming a surface-based temperature inversion leading to a stably stratified lower atmosphere. This evolution is most pronounced during the polar night and gradually weakens during the transition to early spring. Due to shear stress, which comprises a major source of turbulence in the Arctic ABL, a surface mixing layer (SML) can evolve even though stability dampens turbulence (Brooks et al., 2017). The SML relates to the lowermost part of the atmosphere that is turbulent, while the SML height[1] does not necessarily equal the ABL height because disconnected turbulence may occur aloft (Grachev et al., 2013). If low-level clouds form, the surface-based temperature inversion is lifted upward to the cloud top. Then, in addition to mechanically induced turbulence at the surface, a second source of turbulence at the cloud top evolves caused by negative buoyancy due to radiatively cooled air at the cloud top, which leads to a cloud mixed layer (Tjernström and Graversen, 2009; Morrison et al., 2012). In particular, low-level Arctic clouds impact the surface radiative energy budget (Intrieri et al., 2002a; Shupe and Intrieri, 2004) and thus alter the vertical structure of the ABL (Tjernström and Graversen, 2009). Furthermore, the vertical stratification of the ABL influences the formation and longevity of clouds (Intrieri et al., 2002a; Sedlar and Tjernström, 2009; Shupe et al., 2013; Turner et al., 2018), for example, as surface sources of atmospheric moisture, energy, and cloud-forming particles have a significant impact on cloud properties (Gierens et al., 2020; Griesche et al., 2021). While cloudless conditions are rather scarce in the Arctic (Intrieri et al., 2002b), frequently occurring low-level mixed-phase clouds are of major importance for the surface radiative energy budget.

---

[1]In the following parts of this study, we avoid the rather general term ABL height and use the term SML height for our analysis.

Furthermore, local ABL processes might be influenced by the advection of warmer/cooler air masses. As a result, elevated inversions can form, which decouple different layers. These elevated inversions would not meet the classical definition of the ABL, although they may feed back to the air layer near the surface (Mayfield and Fochesatto, 2013). Between the two typical radiative states (cloudless and cloudy), the ABL alternates on the time scale of hours, even though they have not yet been investigated in detail, mostly because corresponding high-resolution profile measurements are missing.

Further issues regarding the turbulent properties and thermodynamic structure of the Arctic ABL in winter include, for example, the heights up to which heat energy is distributed or aerosol particles are mixed from the surface. Therefore, the determination of the SML height is of utmost importance. Widely used approaches to determine the height of the SML are based on observed thermodynamic profiles. An overview of common identification methods can be found in Vickers and Mahrt (2004), Dai et al. (2014), and Jozef et al. (2022). The definition of the top of the SML is not always straight forward; in particular, it becomes even more complex as the criteria are not very pronounced for stable stratification (Mahrt, 1981; Stull, 1988). The basic idea behind the definition of a SML height is that starting from the surface, a property or matter is mixed upwards by turbulence, and this mixing is terminated when the turbulence is no longer strong enough for vertical mixing. An obvious definition of the SML height $h$ is, therefore, based on the vertical distribution of a suitable turbulence parameter and a threshold, which, if below this value, defines the SML height (Dai et al., 2014). Here we use direct balloon-borne turbulence observations, in particular, energy dissipation rate $\varepsilon$ profiles, to estimate the SML height (Balsley et al., 2006). By observing turbulence by in situ measurements, the SML height can be derived directly. One can either define the SML height as the height where $\varepsilon$ drops significantly with height or when the flow is considered non-turbulent based on a "turbulence threshold" (Shupe et al., 2013; Brooks et al., 2017).

Another approach to define the SML height applies the bulk Richardson number $Ri_b$ (Andreas et al., 2000; Zilitinkevich and Baklanov, 2002; Dai et al., 2014; Zhang et al., 2014; Jozef et al., 2022; Peng et al., 2023). $Ri_b$ is derived from the ratio between shear and buoyancy and is a measure of the likelihood of turbulence to exist. A $Ri_b$ below the critical value indicates an atmospheric layer that is likely to remain or become turbulent. Turbulence cannot be sustained, and laminar layers will not become turbulent if the $Ri_b$ is above a critical value. The SML height is defined by the height where turbulence cannot be sustained because the $Ri_b$ exceeds a critical value of $Ri_b$ (Zilitinkevich and Baklanov, 2002; Andreas et al., 2000). This critical value, however, is under discussion and varies, for example, among sites (Vickers and Mahrt, 2004).

We use the turbulence-based SML heights as reference to derive a critical $Ri_b$ for the winter and spring. The in situ turbulence perspective not only allows to derive a critical value but also to evaluate the $Ri_b$ approach. It should be emphasised that this approach can also be applied to a stably stratified atmosphere. Further, continuous, surface-based energy flux measurements can be used to estimate the SML height using Monin-Obukhov similarity theory (Zilitinkevich, 1972; Vickers and Mahrt, 2004). The Monin-Obukhov similarity theory describes the near-surface turbulent exchange processes based on surface measurements. This approach can complement the balloon-borne SML height estimates between balloon launches.

The current study discusses observed profile measurements of Arctic ABL turbulent properties and related effects of clouds on the vertical thermodynamic structure. The data are used to develop a new, more accurate approach to estimate the SML height in Arctic winter and spring. To investigate possible SML height criteria, the in situ energy dissipation rates are compared

to $Ri_b$ profiles and a surface flux-based method. Furthermore, we apply the critical $Ri_b$ value technique for deriving the SML height using both tethered balloon as well as radiosonde data.

## 2 Data and methods

### 2.1 Observations in winter and spring during the MOSAiC expedition

During the MOSAiC expedition, the research vessel (RV) *Polarstern* (Knust, 2017) was frozen to an ice floe drifting from October 2019 until September 2020. The MOSAiC expedition facilitated measurements onboard RV *Polarstern* and on the ice floe covering the atmosphere, Arctic ocean, sea ice, ecosystem, and biogeochemistry throughout an entire seasonal cycle. An overview of the atmospheric measurements is given by Shupe et al. (2022), and information about the sea ice and oceanographic aspects is summarized by Nicolaus et al. (2022) and Rabe et al. (2022). Figure 1a shows the course of the drift of RV *Polarstern* in winter and spring when the tethered balloon *Miss Piggy* (Becker et al., 2020) was deployed from the ice floe close to RV *Polarstern*. A detailed view of the temporal evolution and the balloon flight days is given in Fig. 1b. The balloon was filled with helium, has a volume of about $9\,m^3$, and allows to lift a modular scientific payload of up to $4\,kg$. The tethered balloon enables continuous vertical profiling of the lowermost $1.5\,km$ of the atmosphere with a climbing rate of around $1\,m\,s^{-1}$. Measurements can be performed day and night as well as under cloudy conditions with light icing. The deployment of the balloon is inhibited by strong winds (wind velocity above about $7\,m\,s^{-1}$ at the surface) or precipitation associated with severe icing. Therefore, weather-related selective sampling should be considered for further interpretation.

A hot-wire anemometer package specifically designed for turbulence observations (Egerer et al., 2019), hereafter referred to as the "turbulence probe", was used to measure the data of this study. Data were collected when the ice floe drifted between $86.14°\,N\,122.21°\,E$ and $83.92°\,N\,17.69°\,E$ between 6 December 2019 and 6 May 2020 (Fig. 1b). The entire data set and its processing is described by Akansu et al. (2023b). The instrument is attached to a tether about $10\,m$ to $20\,m$ below the balloon to minimize flow distortions induced by the balloon. In addition, the instrument package attached to the balloon tether is mounted flexibly, it is aligned horizontally and equipped with a tail to keep the setup in the mean flow direction. The instrument consists of a hot wire anemometer to measure wind velocity with $125\,Hz$ temporal resolution, and a thermocouple for air temperature measurements with $10\,Hz$ temporal resolution. Besides the high-frequency records, $1\,Hz$ measurements of wind velocity based on a Pitot static tube have been made, and basic measurements of static pressure, temperature, and relative air humidity. Due to weather conditions, the Pitot tube, which served as a reference for hot wire calibration, was frequently affected by icing. Therefore, a standard meteorology tethersonde, which belongs to the modular balloon equipment and is separated about $10\,m$ from the turbulence probe, was the mean reference for the fast sensors. The turbulence probe was deployed on $34\,days$, sampling the height range from the surface to typically a few hundred meters. After quality control, 99 individual vertical profiles (ascents and descents) have been provided for further analysis. In this study, 81 out of the 99 profiles are used because not all measurements reach the surface, and some contain error-prone temperature data. Figure 1c displays an overview of all temperature profiles, the respective maximum height, and daylight conditions. Table A1 shows all profiles and their launch times.

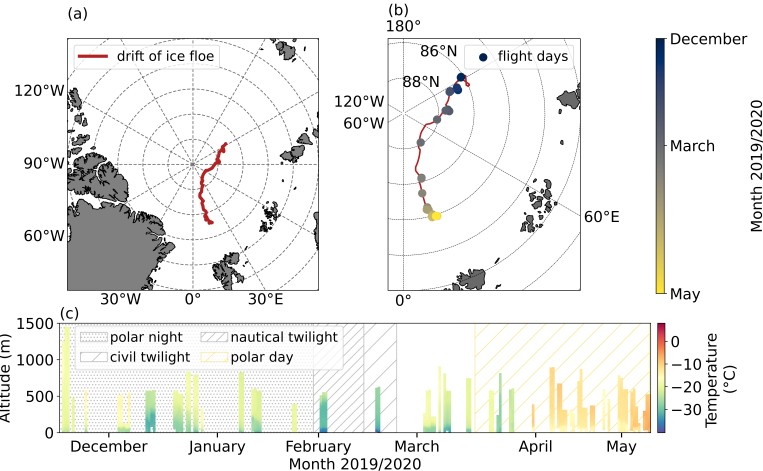

**Figure 1.** Overview of the drift of the ice floe during tethered balloon operations in winter and spring (a), and daylight conditions and location of the days with balloon operations (b). Temperature measurements of all profiles are shown in (c). The background shading indicates the respective daylight conditions during the observation period with the white background indicating the period with a diurnal cycle.

Continuous, near-surface observations of meteorological and turbulence parameters were performed at a location on the ice floe called *Met City* (Shupe et al., 2022). There was about 300 meters distance between *Met City* and the balloon operation site, increasing over time due to ice flow dynamics. Measurements with an ultrasonic anemometer/thermometer were taken at a meteorological tower at 2 m, 6 m, and 10 m heights, serving as surface reference for our balloon observations. Furthermore, the upward ($\uparrow$) and downward ($\downarrow$) broadband terrestrial irradiances $F$ (in units of W m$^{-2}$) were collected at the Atmospheric

Surface Flux Station with a Precision Infrared Radiometer (Shupe et al., 2022). The net terrestrial irradiances $F_{net} = F^{\downarrow} - F^{\uparrow}$ are taken as a proxy for the radiative energy budget at the surface. Furthermore, the measurements at *Met City* include the surface radiometric skin temperature.

    Regular radiosondes were launched every six hours (5, 11, 17, and 23 UTC) from the helicopter deck of RV *Polarstern* (12 m above mean sea level), providing information on the dynamic and thermodynamic structure of the atmosphere. It should

be noted that the data obtained within the first tens of meters can be influenced by the ship itself and should, therefore, be interpreted with caution. The ship can create flow distortions and can serve as a heat island, but also radiosonde data near the surface are often still subject to errors. Especially the wind determined by GPS (Global Positioning System) data might be influenced by the unwinding of the probe from the tether (Achtert et al., 2015; Jozef et al., 2022). To compare the radiosonde and the tethered balloon profile measurements, the corresponding radiosonde data are selected with launch times closest to

the tethered balloon flight. The time difference between both launches is, at most, around 3 hours, in which the SML height typically did not change significantly, as shown by Jozef et al. (2022). However, the atmospheric structure may change within a few hours or even less.

The Arctic ABL can be very shallow, especially in winter and during cloudless conditions. However, also the cloudy Arctic ABL is often shallow, and hence poses challenges on remote sensing approaches to detect the very low cloud layers (Griesche et al., 2020). For the question of the influence of clouds on the ABL dynamics, however, the net irradiances are of main importance, which are directly influenced by the clouds. Therefore, $F_{net}$ measurements at *Met City*, where $F_{net}$ is the cumulative surface irradiance, are used as an independent "cloud indicator". $F_{net}$ can be used to distinguish between cloudless and cloudy conditions during both polar day and night. Cloudless conditions prevail when the net radiation is below -25 W m$^{-2}$, while higher values are associated with clouds (Wendisch et al., 2023a). To avoid ambiguous allocations, data with net radiation in the range between -28 W m$^{-2}$ and -22 W m$^{-2}$ are not considered. Additionally, the cloud condition was manually compared with 360° photographs and total sky imager observations (as far as possible regarding daylight conditions).

## 2.2 Estimation of the surface mixing layer height

In this chapter, we present several methods for determining SML heights based on different available data sets. First, we discuss a method to derive SML heights that benefits directly from local turbulence measurements and will serve as a reference for other approaches in the following (Sect. 2.2.1). A second method is based on the mean temperature and wind speed profiles in the context of the $Ri_b$ criterion (Sect. 2.2.2). Finally, we compare the previous results with SML heights based on the application of the Monin-Obukhov similarity theory and thus on the determination of surface fluxes (Sect. 2.2.3).

### 2.2.1 Direct in situ method

The direct in situ method for estimating the SML is based on the assumption that the turbulence at the top of this layer falls below a certain threshold, indicating the transition from a turbulent to a comparatively laminar layer. For our application, we chose the energy dissipation rate $\varepsilon$ to quantify turbulence because it can be determined as a local parameter from short temporal subsections during a balloon ascent. Based on Kolmogorov's inertial subrange theory, $\varepsilon$ can be estimated in different ways (Wyngaard, 2012). A comparatively robust and proven method has been found to determine $\varepsilon$ using the second-order structure function (Siebert et al., 2006):

$$S^{(2)}(\tau) = \left\langle (u(t+\tau) - u(t))^2 \right\rangle = 2(\varepsilon \cdot U \cdot \tau)^{2/3}, \tag{1}$$

with $u(t)$ the longitudinal wind velocity component as measured at time $t$. The averaging in Eq. (1) denoted by the angle brackets is performed over all $t$, so the structure function $S^{(2)}$ is a function of time lag $\tau$ which has been calculated by applying Taylor's frozen turbulence hypothesis (transferring time lag to spatial increment $r = U \cdot \tau$) with the mean flow velocity $U$ (see Stull (1988) for more details). It has been shown by Frehlich et al. (2003) that 100 samples are sufficient in order to provide robust estimates of local $\varepsilon$. Here, we use an integration over 125 samples (e.g., 1 s) resulting in a vertical resolution of roughly 1 to 2 m.

Profiles of $\varepsilon$ allow direct identification of the SML height, denoted hereafter as $h_\varepsilon$. This method has the disadvantage that there is no physically unambiguous definition of the limit value for $\varepsilon$ and, therefore, different values are used in the literature (Shupe et al., 2013; Brooks et al., 2017), which are close to each other. We tested three values to estimate $h_\varepsilon$ that have been

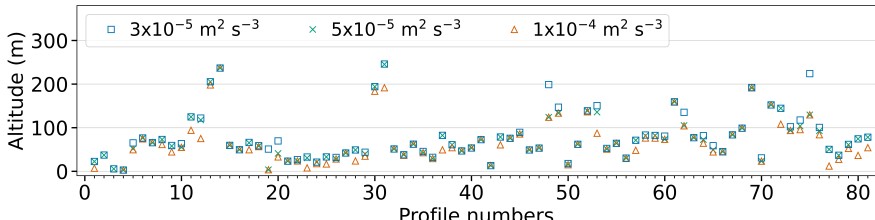

**Figure 2.** Surface mixing layer heights derived from direct turbulence measurements for each profile. Three thresholds separate turbulent and non-turbulent flows: $1x10^{-4}\,\mathrm{m^2\,s^{-3}}$, $5x10^{-5}\,\mathrm{m^2\,s^{-3}}$, and $3x10^{-5}\,\mathrm{m^2\,s^{-3}}$.

used previously: $1x10^{-4}\,\mathrm{m^2\,s^{-3}}$, $5x10^{-5}\,\mathrm{m^2\,s^{-3}}$ and $3x10^{-5}\,\mathrm{m^2\,s^{-3}}$. We consider the SML to be constrained when $\varepsilon$ falls below the threshold for at least 10 consecutive levels, while the $h_\varepsilon$ equals the lowermost of those levels. These 10 levels ensure that the turbulent layer and non-turbulent layer are well separated.

Figure 2 shows the estimates of $h_\varepsilon$ for each profile for the different thresholds. The profile numbers refer to the profiles used in this study. An overview is given in Table A1. The highest threshold almost always yields the lowest $h_\varepsilon$ values, and the
two lowest thresholds are very close and result in nearly identical values of $h_\varepsilon$. To identify the appropriate threshold value, we have compared the SML heights derived by different thresholds with the potential temperature profiles. Additionally, we have examined whether the SML height coincides with a significant change in $\varepsilon$. As a result we have decided on the threshold value of $\varepsilon = 3x10^{-5}\,\mathrm{m^2\,s^{-3}}$.

### 2.2.2  Bulk Richardson number method

The turbulent state of an atmospheric layer can be analyzed using the (gradient) Richardson number $Ri_\mathrm{g}$, which describes the relationship between thermodynamic stability and turbulence-producing horizontal wind shear (e.g., Stull, 1988):

$$Ri_\mathrm{g}(z) = \frac{\frac{g}{\theta}\frac{\partial\theta}{\partial z}}{(\frac{\partial u}{\partial z})^2 + (\frac{\partial v^2}{\partial z})^2}, \tag{2}$$

with the potential temperature of dry air $\theta$, the horizontal wind components $u$ and $v$ (zonal and meridional), the gravitational acceleration $g = 9.81\,\mathrm{m\,s^{-2}}$, and $z$ as height above ground. This stability measure describes whether there is a tendency for
turbulence to weaken or strengthen. $Ri_\mathrm{g}$ smaller than the theoretical value of 0.25 refers to an turbulent atmospheric layer. Therefore, $Ri_\mathrm{g}$ profiles can be used to derive the equilibrium height at which turbulence decays, which coincides with the SML height $h_{Ri_\mathrm{g}}$ (Zilitinkevich and Baklanov, 2002; Andreas et al., 2000). However, the practical use of $Ri_\mathrm{g}$ is somewhat limited because the calculation of (local) wind and temperature gradients based on observational data is often subject to uncertainties that lead to large scatter in the $Ri_\mathrm{g}$ profiles. In particular, the necessary filtering leads to further ambiguities. For this reason,
modified definitions for $Ri_\mathrm{g}$ have been derived, which offer advantages, especially for observational data. An alternative $Ri$ number is the so-called surface bulk Richardson number $Ri_\mathrm{b}$ (Mahrt, 1981; Andreas et al., 2000; Heinemann and Rose, 1990), whose definition is not based on the explicit calculation of local gradients but includes the complete layer from the surface to

the current measurement height $z$:

$$Ri_{\mathrm{b}}(z) = \frac{g \cdot z \cdot \frac{\Delta\theta}{\theta_0}}{U(z)^2}, \tag{3}$$

with $\theta_0$ is the potential temperature of dry air at the surface and $\Delta\theta = \theta(z) - \theta_0$ is the temperature difference between the surface and $z$. According to the basic concept of the surface bulk Richardson number approach, the lower reference level is the surface where the mean horizontal wind velocity equals zero $U(z = z_0)$ and $\theta_0$ is the skin temperature. In the following, $Ri_{\mathrm{b}}$ always refers to the surface bulk Richardson number. Since other studies are often based on radiosonde observations only, the temperature at 2 m is then typically used for $\theta_0$, even though this value may differ from the skin temperature, especially for stably stratified conditions with strong surface temperature gradients. To be consistent with other studies, we first use temperature observations at 2 m from the *Met City* tower for $\theta_0$. Then, for comparison, the same analysis is performed using the skin temperature measured during MOSAiC.

Compared to other bulk $Ri$-definitions, where mean gradients are estimated for distinct layers of $\delta z = 30\,\mathrm{m}$ (Jozef et al., 2022), for example, the $Ri_{\mathrm{b}}$ approach applied here fails if multiple turbulent layers are present. However, as we want to estimate the height of the SML, only the lowermost continuous turbulent layer needs to be detected, and the surface approach with an increasing layer depth is sufficient. Furthermore, from the values of $h_\varepsilon$ presented in Fig. 2, it becomes apparent that the majority of the SML heights lie within the lowermost 100 m, many even do not exceed 50 m altitude and multiple turbulence layers are rare.

The height of the SML $h_{Ri_{\mathrm{b}}}$ is the height at which $Ri_{\mathrm{b}}$ reaches a critical value and the turbulence decays. The top of the SML is, thus, the highest level where $Ri_{\mathrm{b}} < Ri_{\mathrm{bc}}$. However, the definition of $Ri_{\mathrm{bc}}$ is not straightforward and not based on a theoretical concept. It appears that the differences for $Ri_{\mathrm{bc}}$ measured at different sites are larger than the variation within an observation period at a fixed site (Vickers and Mahrt, 2004). When applying 0.25 as a critical value for the bulk approach, the experimentally determined SML heights are overestimated (Brooks et al., 2017). However, this theoretical critical value was derived for the gradient $Ri_{\mathrm{g}}$ number and is therefore not directly applicable to the surface bulk approach. In addition, Eq. (3) is sensitive to the lowest observation level, which may also explain some of the variation in $Ri_{\mathrm{bc}}$. However, using the "reference" SML height $h_\varepsilon$ introduced in Sect. 2.2.1, we can provide a direct and robust estimate for $Ri_{\mathrm{bc}}$ (see Sect. 3.3).

### 2.2.3 Surface flux-based method

If surface energy fluxes are the main drivers for the development of a SML, it is proposed that the SML height is a function of the Monin-Obukhov length $L$ (Vickers and Mahrt, 2004):

$$L = -\frac{u_\star^3}{B_{\mathrm{s}}}, \tag{4}$$

with $B_{\mathrm{s}} = g \cdot T_0^{-1} \langle w' \cdot T' \rangle$ being the surface buoyancy flux and $u_\star^2 = -\langle u' \cdot w' \rangle$ being the friction velocity. Besides more simple relationships (e.g., $h \propto L$, Kitaigorodskii, 1960), Zilitinkevich (1972) proposed a formulation including the Earth's

rotation:

$$h_{\mathrm{MO}} = C\sqrt{\frac{u_\star L}{f}}, \tag{5}$$

with $C$ being a scaling constant of $\mathcal{O}(1)$ and the Coriolis parameter $f$. All parameters included in Eq. (5) were calculated continuously using ultrasonic anemometer readings at the *Met City* tower. With the Monin-Obukhov scaling method, referred to as the MO method hereafter, the SML height can be derived continuously and complements balloon-borne SML height estimates. However, Eq. (5) is only valid for stable stratification ($B_s < 0$ and $L > 0$), and the method fails if further sources of turbulence are present at higher levels (such as clouds or Low-Level Jets (LLJs)). Again, the in situ turbulence method is

helpful to assess the applicability of the MO method (Sect. 3.5).

## 3  Results and discussion

### 3.1  Surface-based versus elevated inversion

In the introduction, we have argued that the Arctic winter ABL mainly alternates between cloudless and cloudy conditions. We begin at this point by illustrating these two typical conditions presenting two examples. The first case represents a cloudless

ABL with a pronounced surface inversion, and the second case describes a cloudy ABL and the resulting elevated inversion at the cloud top with a well-mixed layer below the inversion. Figure 3 shows the two example measurements of potential temperature $\theta$, horizontal wind velocity $U$, local energy dissipation rate $\varepsilon$, and surface bulk Richardson number $Ri_b$ as a function of altitude. Cloudless conditions prevailed during a profile observed on 5 March 2020 with a strong surface-based temperature inversion ($\Delta\theta \approx 7\,\mathrm{K}$ within about 40 m) up to 50 m followed by a less stably stratified layer above (Fig. 3a). The

wind velocity $U$ increases with height from the surface up to a height of about 50 m, and then remains almost constant up to the maximum height of the profile (Fig. 3b). The strongest increase in $U$ is in the lowest altitude layers up to about 30 m, the region with the highest turbulence in terms of $\varepsilon$ (Fig. 3c). Above 30 m, the turbulence decreases rapidly by two to three orders of magnitude and according to the threshold, as defined in Sect. 2.2.1, the upper limit of the SML height is reached here. The $Ri_b$ (Fig. 3d) increases almost linearly with height for the cloudless conditions.

The example for the vertical stratification under cloudy conditions, as measured on 29 December 2019, shows an elevated inversion with its base at around 220 m. A well-mixed layer prevails below the inversion, with $U$ gradually increasing from the surface to the inversion base from $6\,\mathrm{m\,s^{-1}}$ to $9\,\mathrm{m\,s^{-1}}$. Clearly, the turbulence reaches from the surface up to about 240 m height, where $\varepsilon$ rapidly drops below the threshold, indicating the SML height. The extent of the SML is also clearly visible in the $Ri_b$ profile (Fig. 3d). The $Ri_b$ remains almost constant with height in this region with values close to zero, it starts to

increase significantly only at the inversion base height, exactly where $\varepsilon$ starts to decrease quite abruptly and the upper limit of the SML height is reached.

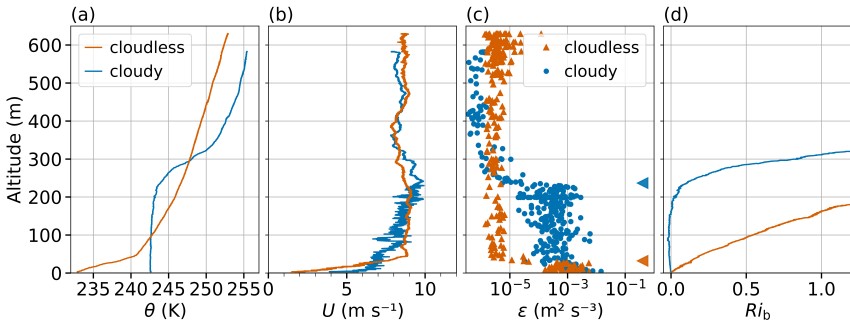

**Figure 3.** Tethered balloon-borne profiles obtained under cloudless conditions (orange) on 5 March 2020, 12:27 UTC, and under cloudy conditions (blue) on 29 December 2019, 11:55 UTC. Panel (a) shows the profiles of potential temperature over height, (b) the wind velocity, (c) the derived energy dissipation rates and (d) the surface bulk Richardson number. The SML heights for cloudless (orange) and cloudy conditions (blue) derived by in situ turbulence records are indicated by left pointing triangles on the right in (c).

## 3.2 Vertical mean and turbulent structure of the ABL

To illustrate the main features of the mean temperature stratification for the two typical situations, we normalize, plot, and average all measured profiles (Fig. 4), distinguishing between surface (Fig. 4a) and elevated (Fig. 4b) inversions. The height is normalized by $h_\varepsilon$, and the temperature is shifted by the surface value $\theta_0$ and normalized by the inversion strength [$\Delta\theta = \theta(h_\varepsilon) - \theta_0$], so that the normalized temperature in the SML height takes the value one. A similar plot, including the contrasting summertime Arctic, can be found in Tjernström and Graversen (2009).

Due to operational constraints, not all profiles reach the same height, but all launches used here exceed 200 m altitude, and therefore most profiles exceed at least twice $h_\varepsilon$. The two averaged temperature profiles have quite different characteristics. While the profile is nearly linear from the surface to $h_\varepsilon$ for cloudless conditions, the well-mixed sub-cloud layer for the cloudy case shows a gradual increase in temperature, rising much more sharply at $h_\varepsilon$. As is often observed in Arctic clouds, the temperature increase at the inversion base already begins inside the cloud; this normalization does not consider the cloud layer thickness, and hence, the temperature curve in Fig. 4b must be interpreted with caution. In addition, it must be emphasized that we do not present the equivalent potential temperature, which under adiabatic conditions is also height constant within the cloud.

Figure 5 shows the relative probability distribution of $\varepsilon$ using 5 m height intervals of all measured profiles. The turbulence distribution for surface-based inversions (Fig. 5a, 30 profiles) shows the highest values near the surface (wind shear-driven turbulence) and significantly lower ones above approximately 60 m height. Higher probabilities of $\varepsilon$ at around 500 m height occurred during single events and might be related to LLJs. However, the observations are too sparse to draw conclusions about the occurrence of possible multi-layer turbulent structures. For ABL structures with elevated inversions (Fig. 5b, 34 profiles), the turbulence is still highest in the lowermost tens of meters. But here, the turbulence reaches up to around 200 m. The chosen threshold of $\varepsilon = 3\text{x}10^{-5}\,\text{m}^2\,\text{s}^{-3}$ distinguishes turbulent and non-turbulent regions of the profiles. With this threshold

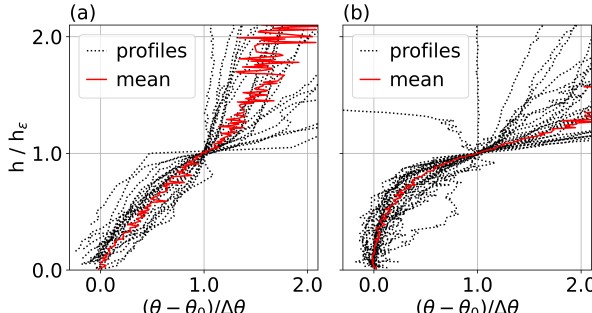

**Figure 4.** Normalized potential temperature profiles of tethered balloon-borne measurements over height. All profiles are separated in respect to the inversion heights, such as surface-based inversion (a) or elevated inversion (b). The height is normalized using the turbulence-based SML height $h_\varepsilon$ while the temperature is normalized using the potential temperature gradient of the SML and the potential temperature close to the surface.

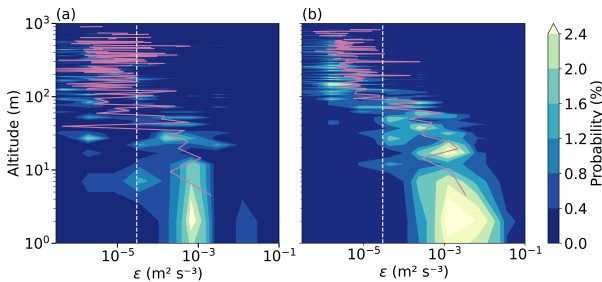

**Figure 5.** Probability of the energy dissipation rates as a function of height. The profiles are distinguished in respect to the temperature profile: (a) shows all profiles with surface-based inversions and (b) profiles with elevated inversions. The probability of $\varepsilon$ is calculated for 5 m height bins starting at a height of 2 m. To complement the data from the surface, the first bin reaches from the surface up to a height of 2 m. Additionally, the median (solid line) profile of $\varepsilon$ (5 m bins starting at 2 m height) is given in purple. The turbulence threshold value of $3\mathrm{x}10^{-5}\,\mathrm{m^2\,s^{-3}}$ is depicted by the white vertical dashed line in both panels.

approach, almost all observations above 300 m height for elevated inversions (60 m height for surface inversions) are considered non-turbulent and thus well above the SML. In contrast to the summertime Arctic ABL, where Brooks et al. (2017) found two separated turbulence maxima indicating decoupling, we did not identify clearly pronounced turbulence layering, and decoupling plays a minor role.

### 3.3 Estimating the critical Bulk Richardson Number

Estimating the SML height based on the $Ri_\mathrm{b}$ requires a critical value $Ri_\mathrm{bc}$. Here, we apply a method used by Vickers and Mahrt (2004) by plotting the buoyancy term against the shear term for all profiles at $h_\varepsilon$ in Fig. 6. The slope of the linear fit

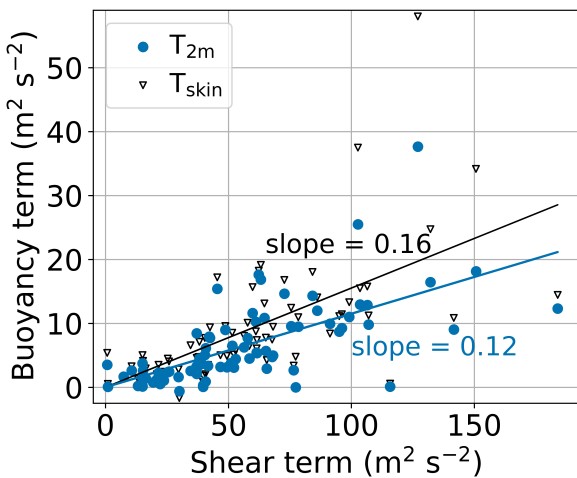

**Figure 6.** The buoyancy term versus shear term of the surface bulk Richardson number calculated according to Eq. 3 using the temperature at 2 m height $T_{2\,m}$ (blue) and the skin temperature $T_{skin}$ (black) as $\theta_0$. The solid line indicates the least squares fit that is forced through the origin. The critical bulk Richardson number $Ri_{bc}$ is equivalent to the slope of the fit. The critical values are 0.12 for $T_{2\,m}$ and 0.16 for $T_{skin}$.

corresponds to the critical surface bulk Richardson number $Ri_{bc}$. Based on data from 80 profiles, we derive a critical value of $Ri_{bc} = 0.12$, about half the theoretical value of 0.25. As mentioned in Sect. 2.2.2, we would like to point out again that this "theoretical value" should be understood as an order of magnitude for the $Ri_b$ approach rather than a reference value. Applying the same analysis with the skin temperature as $\theta_0$, we derive $Ri_{bc} = 0.16$ (Fig. 6). This difference shows clearly how strongly the derived value for $Ri_{bc}$ depends on the selected surface reference. Furthermore, we can assume that the temperature measured

at a height of 2 m on a mast is also significantly more accurate than a corresponding measurement with a radiosonde, which has comparatively large inaccuracies in the lower ranges.

     To obtain a rough measure of the robustness of the $Ri_{bc}$ estimate, Fig. 7 shows the frequency of occurrence of $Ri_{bc}$. Clearly, the majority of $Ri_{bc}$ using the temperature at 2 m height as $\theta_0$ is centered around the critical value of 0.12, with a few outliers exceeding this value. The frequency distribution of $Ri_{bc}$ with $T_{skin}$ as $\theta_0$ is slightly shifted towards higher $Ri_{bc}$ values.

As described by Vickers and Mahrt (2004), the $Ri_{bc}$ not only varies between sites but is also a function of atmospheric conditions (cloudless, cloudy). To understand which influence may lead to different values of $Ri_{bc}$, we distinguished the data by the cloud conditions using $F_{net}$. While we have derived $Ri_{bc}$ for cloudless and cloudy conditions, the differences between the two typical ABL types are negligible. The critical value of $Ri_{bc} = 0.12$ can be used for all conditions (with temperature measurements at 2 m height as $\theta_0$).

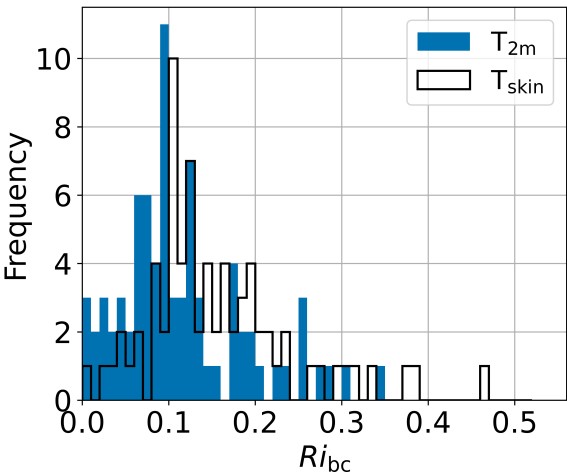

**Figure 7.** Frequency distributions of surface bulk Richardson numbers at the SML height $h_\varepsilon$ using the temperature at 2 m height $T_{2\,\mathrm{m}}$ (blue) and the skin temperature $T_{\mathrm{skin}}$ (black) as $\theta_0$.

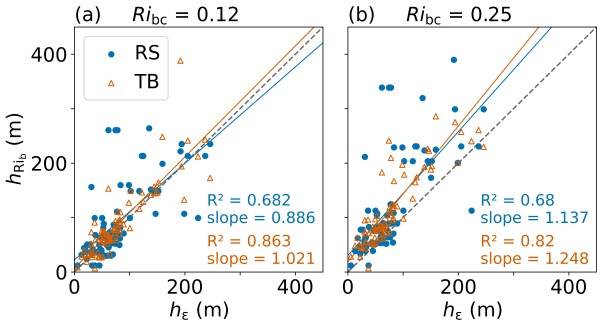

**Figure 8.** SML height comparisons between the $Ri_b$ approach $h_{Ri_b}$ and turbulence $h_\varepsilon$ based on tethered balloon (TB, orange) and radiosonde (RS, blue) profiles. For both data sets, two different critical $Ri_b$ are used: 0.12 (a), and 0.25 (b). The dashed line represents the 1:1 line. Solid lines are the least squares fits for tethered balloon (orange) and radiosonde (blue) profiles; $R^2$ and slope values are given for each fit.

### 3.4 Surface mixing layer height estimates based on a mean critical Richardson number

To assess the impact of using an averaged critical $Ri_b$ number on the individual SML height, we compare $h_{Ri_b}$ to the turbulence-based value $h_\varepsilon$. In addition to the tethered balloon profiles, we apply the same $Ri_b$ method to the radiosondes profiles using the *Met City* tower data as a surface reference. Figure 8 shows the comparison of the SML heights applying two different $Ri_{bc}$ for both tethered balloon-borne as well as radiosonde profiles. We use those radiosonde profiles launched closest to the tethered balloon flight.

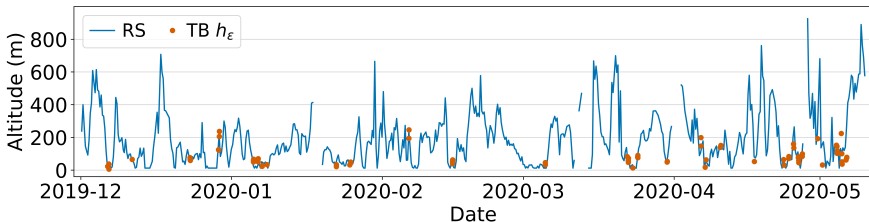

**Figure 9.** Time series of SML heights from radiosonde data derived with surface bulk Richardson number method using $Ri_{bc} = 0.12$. The time series of radiosonde launches is displayed between 1 December 2019 and 10 May 2020 (blue). Additionally, the SML heights $h_\varepsilon$ of tethered balloon-borne profiles are shown as orange dots.

For the tethered balloon observations, the slope using the theoretical value of 0.25 is about 22 % higher than when using $Ri_{bc} = 0.12$. As all intercepts are positive, a slope $> 1$ indicates an overestimation of $h$. Compared to $h_\varepsilon$, $h_{Ri_b}$ with $Ri_{bc} = 0.12$ is about 15 % higher while $Ri_{bc} = 0.25$ leads to about 60 % overestimation of $h$. This supports the need for an accurate estimate of $Ri_{bc}$. If we apply the newly determined $Ri_{bc}$ and the theoretical value for comparison to the radiosonde observations, the inaccuracies in $h$ become even more apparent. Besides the slope, also the intercept increases (from 22 with $Ri_{bc} = 0.12$ to 30 with $Ri_{bc} = 0.25$), leading to an overestimation of $h$. It should be noted that the reference height $h_\varepsilon$ was not determined at the same time as the height based on the radiosonde ascents, which may explain some of the observed differences.

Furthermore, we use $Ri_{bc} = 0.12$ to derive $h$ for all radiosonde launches collected during winter and spring. Figure 9 shows the time series of the derived SML height and additionally $h_\varepsilon$ of the tethered balloon-borne turbulence estimates. In Fig. 9, a high variability of the SML height is displayed during the entire observation period, reaching from a few tens of meters up to 600 m and more (maximum 925 m). It has to be considered that the minimum detection limit of the radiosonde is of about 12 m due to launching from the helicopter deck. A significant growth of the SML is likely related to weather or storm events, while the SML is shallower for calm or stable conditions. The vast majority (around 81 %) of the SML heights derived from the radiosondes during this period are below 300 m, and around half of the profiles contained an SML with height less than 150 m. The tethered balloon operations primarily cover periods of lower wind velocities and shallower SML heights, and during storm events the SML can be much deeper. Whether $Ri_{bc}$ may change during these events remains open and can not be answered in this study.

Frequency distributions of all derived SML heights using radiosonde data are shown in Fig. 10. During the period considered in this plot, the majority of the SML heights are below 250 m (Fig. 10a). Separating between cloudless and cloudy conditions shows that SML heights are spread rather equally for cloudy conditions but are significantly smaller for cloudless conditions (Fig. 10b). During daylight conditions (Fig. 10c and d), the lowest SML heights are measured under cloudless conditions. However, the number of daylight profiles is lower than the number of night profiles for our analysis period. Distributions of the SML heights during polar night are shown in Fig. 10e for all conditions, and separated by cloud conditions in Fig. 10f. We see that the majority of cloudless conditions lead to SML heights of a maximum of 100 to 200 m, while cloudy conditions do not

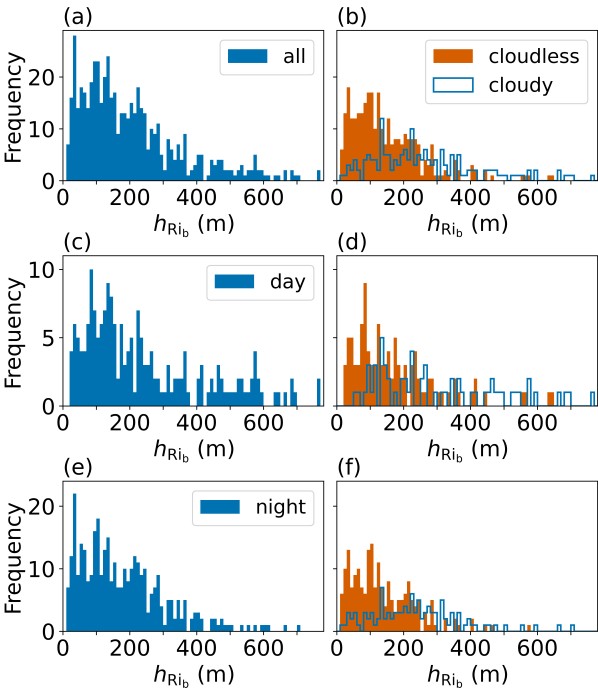

**Figure 10.** Frequency distribution of SML heights using radiosonde data between 1 December 2019 and 10 May 2020. The radiosonde SML heights are based on $Ri_b$ approach with $Ri_{bc} = 0.12$. The frequency distributions are given for (a) all radiosonde profiles, (b) all data separated according to cloud conditions, (c) all profiles with daylight conditions, and (e) all profiles during night, (d) and (f) show the cloudless and cloudy conditions in orange and blue, respectively.

show any tendency towards a distinct SML height. This distribution shows the variety of clouds and their respective influence on the SML. Furthermore, the wind velocity can play a role in the variation of the SML heights as the wind velocity can vary significantly during cloudy conditions.

### 3.5 Surface mixing layer height estimates based on Monin-Obhukov scaling

Under stable conditions ($B_s < 0$ and $L > 0$), we apply the surface flux-based method introduced in Sect. 2.2.3 to determine the SML height $h_{MO}$ based on Monin-Obukhov scaling. The two covariances underlying the definition of $L$ are estimated from ultrasonic anemometer/thermometer observations from the *Met City* tower at 2 m height. A running average ($\tilde{x}(t)$) with a centered window over 5 minutes is applied to define the time series of the fluctuating part as $x'(t) = x(t) - \tilde{x}(t)$ where $\langle x'(t) \rangle = 0$ is fulfilled and $x$ is one of the velocity components ($u, v, w$) or virtual temperature $T_v$ (actually, an ultrasonic thermometer measures the so-called "sonic temperature" which differs only very slightly from the virtual temperature at low humidity values). The averaging to calculate the kinematic fluxes leading to $L$ and, finally, to $h_{MO}$ is done over 30 minutes centered around the balloon ascents/descents.

Figure 11 shows $h_{\mathrm{MO}}$ including $h_\varepsilon$ as reference (Fig. 11a) and the net radiation $F_{\mathrm{net}}$ (Fig. 11b) with a threshold of $-25\,\mathrm{W\,m^{-2}}$ separating cloudless from cloudy conditions. This plot shows a qualitative agreement between $h_{\mathrm{MO}}$ and $h_\varepsilon$ for cases with two or more subsequent values with $F_{\mathrm{net}} < -25\,\mathrm{W\,m^{-2}}$ covering longer-lasting cloudless cases. That is particularly true for the polar

night and twilight periods. However, $h_{\mathrm{MO}}$ is often slightly lower than $h_\varepsilon$. To determine the scaling height, we averaged over 30 minutes, thus comparing 30-minute statistics with a "snapshot" of the atmosphere. We have tested shorter averaging periods (5 minutes) for profiles 6, 15, and 48. However, this does not explain all discrepancies between the two heights, so we assume that the variations between $h_{\mathrm{MO}}$ and $h_\varepsilon$ are more likely to indicate atmospheric conditions. Nevertheless, $h_{\mathrm{MO}}$ represents the SML height well during wintertime and cloudless conditions.

In contrast, for profiles 11 to 14 on 29 December 2019 (period I in Fig. 11, cf. Fig. 3), we find $F_{\mathrm{net}} > -25\,\mathrm{W\,m^{-2}}$ describing a more cloudy situation with increasing $F_{\mathrm{net}}$. At the beginning of this cloudy period, we observe a reasonable agreement between $h_{\mathrm{MO}}$ and $h_\varepsilon$, but eventually, with further increasing $F_{\mathrm{net}}$, $L$ becomes negative, and MO theory fails to predict an SML height. Therefore, with rapidly changing cloud cover, SML height determinations using MO theory are only partially successful. $F_{\mathrm{net}}$ observations can at least indicate these possible problems if no vertical profiles of thermodynamic or turbulent parameters are

available.

During a short period on 6 February 2020 (profiles 30 and 31, period II), a situation with an apparent disagreement between $h_{\mathrm{MO}}$ and $h_\varepsilon$ but $F_{\mathrm{net}} < -25\,\mathrm{W\,m^{-2}}$ and $L$ being positive has been identified. This cloudless period was influenced by an LLJ, which added another source of turbulent kinetic energy well above the surface. The resulting profile of $\varepsilon$, therefore, never falls below the threshold for $h_\varepsilon$ between the core of the LLJ and the surface, indicating a much higher SML height than $h_{\mathrm{MO}}$.

With the beginning of the twilight (15 February 2020, profiles 34 to 36, period III), the cloudless ABL was characterized by a surface inversion and an elevated inversion above, separated by a weakly stable to neutral stratified layer in between, as frequently observed in high latitudes during winter (Mauritsen, 2007). Below the elevated inversion, $\varepsilon$ was always slightly above the threshold for $h_\varepsilon$, suggesting at least some vertical turbulent mixing. However, $h_{\mathrm{MO}}$ significantly underestimates the SML height under those conditions.

In the context of clouds, another particular situation may cause misinterpretation. Shallow ground fog within a neutrally stratified SML up to about $45\,\mathrm{m}$ capped by an inversion was observed in late March (profiles 46 to 51, period IV). The fog was optically quite thin, associated with $F_{\mathrm{net}} < -25\,\mathrm{W\,m^{-2}}$. Due to the elevated inversion, the turbulence threshold was exceeded below $100\,\mathrm{m}$. MO theory failed to predict an SML height because $L < 0$.

The situation becomes even more complicated in the transition to the polar day when the thermal stratification changes from

rather stable to neutral or unstable, and the cloud situation - even with some foggy days - often becomes very complex. These conditions pose challenges for the MO method, especially at the end of the observation period when $L$ is often negative, and thus no SML height can be determined.

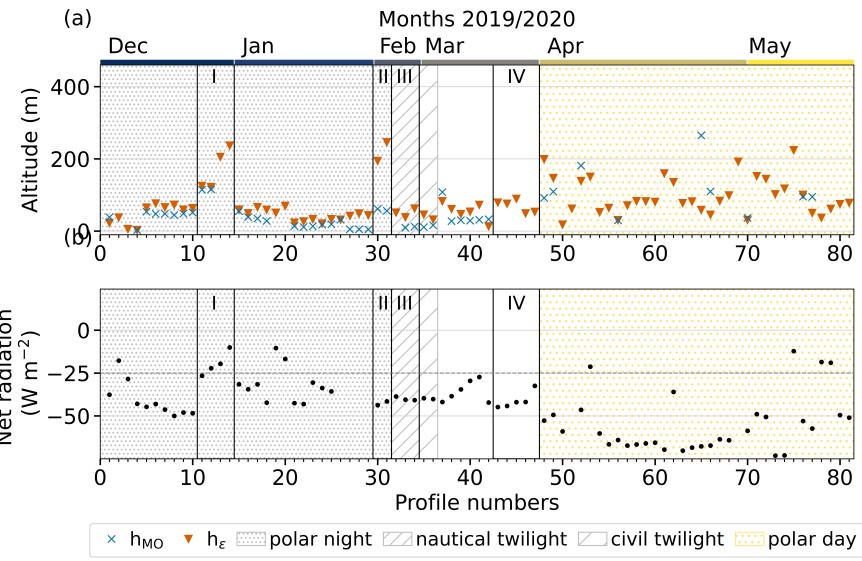

**Figure 11.** SML heights (a) estimated based on MO method ($h_{\mathrm{MO}}$) and direct in situ turbulence profiles ($h_\varepsilon$) as a function of profile number, and (b) the net radiation $F_{\mathrm{net}}$ as observed at the surface. Additionally to the profile numbers, the respective month of the year is indicated at the top axis in (a). Further, shading refers to daylight conditions similar to Fig. 1. Periods discussed in the text are marked by bounding and enumerated with roman numbers.

## 4  Summary and conclusion

The main focus of this study is the evaluation of different methods for determining the surface mixing layer (SML) height for typical conditions observed in the central Arctic during MOSAiC in winter and early spring. The two typical observed conditions - cloudless with a near-surface temperature inversion and cloudy with an elevated inversion at the cloud top - were analyzed in detail. Further, the applicability of the different SML height determinations was investigated for both conditions individually since they offer partly fundamentally different preconditions.

A major advantage of our dataset is the high-resolution turbulence measurements, which allow direct estimations of the SML height under the basic assumption that the turbulent mixing originating from the surface ends at the height where the turbulence falls below a certain threshold, and the flow thus becomes quasi-laminar. Since this transition from a turbulent to an almost laminar layer occurs quite suddenly, this method is relatively robust concerning the choice of the threshold. This method is not based on any further assumptions and, thus, is used here as a reference method.

This reference height was used to apply the surface bulk Richardson number method and to derive the critical value $Ri_{\mathrm{bc}}$ for the conditions observed during the winter and spring of MOSAiC. It was found that an average value of $Ri_{\mathrm{bc}} = 0.12$ can be recommended. We also derived $Ri_{\mathrm{bc}}$ individually for the two ABL conditions (cloudless and cloudy). The differences between these values are minimal (about 7 % deviation) and, therefore, negligible. That we did not observe a difference in the $Ri_{\mathrm{bc}}$ for

the two cases may be somewhat surprising at first glance since the sources of turbulence are different. For the cloudless case, we have mainly the shear-induced turbulence at the surface, whereas for the cloudy conditions, turbulence from the cloud top is added, and both overlap. In addition, surface cooling, and hence stability, is reduced in the presence of clouds, leading to less suppressed wind-shear-driven turbulence. It should be taken into account that we have only determined one term of the energy balance equation with the measured energy dissipation, and, therefore, we cannot make any further statement about the spatial distribution of the turbulent kinetic energy, its sources, or vertical transport. This analysis would require much more complex measurements, including the precise three-dimensional wind vector for covariance measurements, which was not the intention of this work. As long as the conditions as described by the (bulk) Richardson number are sufficient for the existence (or generation/suppression) of turbulence ($Ri_b < Ri_{bc}$), the cause for turbulence generation is not of concern for the current study. Thus, the independence of the observed $Ri_{bc}$ from turbulence generation in both cases (cloudy and cloudless ABL) is physically reasonable.

One of the main advantages of this work is that the $Ri_b$ method can be applied directly to the regularly performed radiosonde ascents with the determined value $Ri_{bc} = 0.12$. This approach provides reliable estimates of the SML height that are about 40 % lower compared to using the canonical value of $Ri_{bc} = 0.25$ ($\langle h_{Ri_{bc}=0.25}/h_{Ri_{bc}=0.12}\rangle = 1.407$). This advantage is particularly important when turbulence measurements from the balloon are not available due to weather or other reasons. However, the $Ri_b$ method requires skin temperature measurements or reliable temperature measurements at a height of 2 m.

There are, of course, other limitations to this study that need to be considered when interpreting the results. For example, we were not able to quantify coupled versus decoupled clouds, which are often observed during the Arctic summer and could change the results. This raises the question of whether the often weak inversion, decoupling the sub-cloud layer from the surface layer, is sufficient to push the turbulence below the threshold value and thus to define the SML height and the height at which $Ri_{bc}$ is determined. The answer to this question urgently requires comparable measurements in the Arctic summer, where these cases are more frequently observed. A valuable supplement to understanding the influence of coupling/decoupling is in situ cloud observations on the balloon.

Since the balloon-borne turbulence profiles are not continuously available, this does not allow for an adequate investigation of the evolution of the SML height. For this reason, we investigated the applicability of the Monin-Obukhov scaling (MO) theory to estimate the SML height. Applying MO to near-surface turbulent flux measurements from tower-based observations, SML heights can be estimated under certain assumptions and compared with the reference heights. Especially for the wintertime, when stable and cloudless conditions prevail, the MO method nicely complements the ABL characterization and provides reliable SML heights. Since the MO method is not applicable for additional sources of turbulence above the surface, such as cloud top cooling or wind shear caused by LLJs, these limitations must be excluded by appropriate observations. Here, at least additional radiosondes are essential for an assessment, while determining surface fluxes alone would often lead to erroneous estimates of the SML height for our observations.

Another point to be considered is the sensitivity of $Ri_{bc}$ on the selection of the lowest observation level. While the mean $Ri_{bc}$ is 0.12 in this study for a 2 m temperature as the surface reference level, $Ri_{bc}$ increases to 0.16 when the skin temperature is

used as the surface level temperature. Therefore, values taken from the literature should always be interpreted with the specifics of the applied approach taken into account.

Since $Ri_{bc}$ does not seem to be universal for all field experiments and we do not yet know the exact cause for the sometimes considerable variability described in the literature, it will probably be necessary in the future either to determine $Ri_{bc}$ individually for each experiment or to measure the SML height directly using turbulence measurements.

Unfortunately, our observations cannot answer other important questions, such as the temporal evolution of the SML height during the transition from cloudless to cloudy conditions and vice versa, because we do not have continuous observations during such an ABL evolution. A height estimation using the MO method is no longer applicable here due to the influence of the cloud. Furthermore, the time gap between consecutive balloon profiles was too large to observe this transition in detail. The latter holds for the tethered balloon turbulence profiles and even more for the radiosonde observations. Ultimately, we believe that only continuous turbulence profile measurements with an optimized measurement strategy can help answer such questions.

Finally, we come to the conclusion that turbulence profile measurements are the most reliable method to determine the SML height, but profile measurements with radiosondes can also be useful to either determine the SML height using the Richardson approach or to check the boundary conditions in combination with MO theory.

*Data availability.* All tethered balloon-borne data (Akansu et al., 2023a) and radiosonde data (Maturilli et al., 2022), as well as the 360° photographs (Engelmann et al., 2022, 2023a, b) and total sky imager observations (Nicolaus et al., 2021) are available on PANGAEA. Surface-based tower measurements are available on the Arctic Data Center (Cox et al., 2023).

*Author contributions.* E.A. processed and analyzed the data and prepared the manuscript with contributions from all co-authors. H.S. designed the turbulence probe. H.S., S.D., and M.W. contributed to the data analysis and scientific discussion.

*Competing interests.* The authors declare that they have no conflict of interest.

*Acknowledgements.* We gratefully acknowledge the funding by the Deutsche Forschungsgemeinschaft (DFG, German Research Foundation) – project number 268020496 – TRR 172, within the Transregional Collaborative Research Center "ArctiC Amplification: Climate Relevant Atmospheric and SurfaCe Processes, and Feedback Mechanisms (AC)[3] " in sub project A02. Data used in this manuscript were produced as part of the international Multidisciplinary drifting Observatory for the Study of the Arctic Climate (MOSAiC) with the tag MOSAiC20192020 and the Project_ID: AWI_PS122_01, AWI_PS122_02, AWI_PS122_03. Here, our special thanks go to A. Sommerfeld, J. Gräser, and R. Jaiser from the Alfred Wegener Institute, who performed the balloon measurements during the first three legs. Tower-based measurements were made by the University of Colorado / NOAA surface flux team supported by the National Science Foundation (OPP-1724551). Surface

radiation measurements were provided by the Atmospheric Radiation Measurement (ARM) User Facility, a U.S. Department of Energy (DOE) Office of Science User Facility Managed by the Biological and Environmental Research Program. Radiosonde data were obtained through a partnership between the leading Alfred Wegener Institute (AWI), the atmospheric radiation measurement (ARM) user facility, a U.S. Department of Energy facility managed by the Biological and Environmental Research Program, and the German Weather Service (DWD). Finally, we thank O. Persson and M. Shupe for fruitful and stimulating discussions on the Arctic atmospheric boundary layer.

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

# Appendix A: Appendix A

**Table A1.** Overview of tethered balloon-borne profiles. The profile number, start time, and derived SML heights are given using the direct in situ method ($h_\varepsilon$), the bulk Richardson number approach with the surface temperature at 2 m height and a critical value of $Ri_{bc} = 0.12$ ($h_{Ri_b}$), and the MO method ($h_{MO}$).

| # | Time (UTC) | $h_\varepsilon$ (m) | $h_{Ri_b}$ (m) | $h_{MO}$ (m) | # | Time (UTC) | $h_\varepsilon$ (m) | $h_{Ri_b}$ (m) | $h_{MO}$ (m) |
|---|---|---|---|---|---|---|---|---|---|
| 1 | 2019-12-06 12:18:29 | 22 | 19 | 38 | 42 | 2020-03-23 12:17:35 | 13 | 28 | 32 |
| 2 | 2019-12-06 18:15:29 | 37 | 7 | | 43 | 2020-03-24 11:44:20 | 79 | 81 | |
| 3 | 2019-12-06 18:57:32 | 6 | 8 | | 44 | 2020-03-24 12:47:28 | 76 | 86 | |
| 4 | 2019-12-07 06:12:50 | 3 | 8 | 3 | 45 | 2020-03-24 13:12:10 | 89 | 88 | |
| 5 | 2019-12-11 13:27:46 | 65 | 59 | 54 | 46 | 2020-03-30 13:28:35 | 49 | 54 | |
| 6 | 2019-12-23 11:10:27 | 76 | 59 | 47 | 47 | 2020-03-30 13:56:55 | 53 | 56 | |
| 7 | 2019-12-23 11:30:21 | 66 | 61 | 48 | 48 | 2020-04-06 12:21:27 | 199 | 133 | 92 |
| 8 | 2019-12-23 11:50:07 | 73 | 72 | 44 | 49 | 2020-04-06 12:47:00 | 147 | 145 | 109 |
| 9 | 2019-12-23 12:41:36 | 59 | 52 | 48 | 50 | 2020-04-07 09:01:10 | 18 | 40 | |
| 10 | 2019-12-23 12:54:10 | 64 | 57 | 51 | 51 | 2020-04-07 14:09:33 | 62 | 68 | |
| 11 | 2019-12-29 07:50:18 | 125 | 129 | 114 | 52 | 2020-04-10 11:33:16 | 139 | 142 | 181 |
| 12 | 2019-12-29 08:05:42 | 122 | 146 | 115 | 53 | 2020-04-10 14:26:00 | 150 | 146 | |
| 13 | 2019-12-29 11:27:41 | 205 | 241 | | 54 | 2020-04-17 11:55:42 | 52 | 74 | |
| 14 | 2019-12-29 11:55:41 | 237 | 243 | | 55 | 2020-04-23 12:31:52 | 64 | 34 | |
| 15 | 2020-01-05 11:51:44 | 59 | 56 | 55 | 56 | 2020-04-23 14:27:10 | 30 | 52 | 30 |
| 16 | 2020-01-05 12:19:12 | 49 | 75 | 39 | 57 | 2020-04-24 13:39:04 | 71 | 79 | |
| 17 | 2020-01-05 13:01:29 | 66 | 72 | 35 | 58 | 2020-04-24 14:18:55 | 83 | 89 | |
| 18 | 2020-01-05 13:17:17 | 59 | 70 | 28 | 59 | 2020-04-24 14:47:16 | 82 | 96 | |
| 19 | 2020-01-06 10:56:53 | 51 | 29 | | 60 | 2020-04-24 15:00:08 | 81 | 95 | |
| 20 | 2020-01-06 11:22:02 | 70 | 62 | | 61 | 2020-04-25 12:58:12 | 159 | 248 | |
| 21 | 2020-01-07 07:19:21 | 23 | 28 | 14 | 62 | 2020-04-25 14:41:58 | 135 | | |
| 22 | 2020-01-07 07:44:02 | 27 | 33 | 11 | 63 | 2020-04-26 12:20:09 | 77 | 89 | |
| 23 | 2020-01-08 07:25:12 | 33 | 36 | 13 | 64 | 2020-04-26 14:20:25 | 82 | 70 | |
| 24 | 2020-01-22 12:59:45 | 21 | 32 | 18 | 65 | 2020-04-26 14:44:57 | 59 | 62 | 265 |
| 25 | 2020-01-22 13:17:39 | 33 | 32 | 19 | 66 | 2020-04-26 14:56:27 | 45 | 60 | 109 |
| 26 | 2020-01-25 07:23:57 | 32 | 44 | 31 | 67 | 2020-04-27 08:30:07 | 84 | 110 | |
| 27 | 2020-01-25 11:06:54 | 42 | 36 | 5 | 68 | 2020-04-27 08:51:28 | 99 | 127 | |
| 28 | 2020-01-25 12:06:00 | 49 | 27 | 5 | 69 | 2020-04-30 12:19:58 | 192 | 388 | |
| 29 | 2020-01-25 12:24:59 | 44 | 28 | 5 | 70 | 2020-05-01 12:06:06 | 31 | 73 | 36 |
| 30 | 2020-02-06 11:19:46 | 194 | 194 | 61 | 71 | 2020-05-04 08:40:34 | 152 | 164 | |
| 31 | 2020-02-06 11:33:48 | 246 | 173 | 56 | 72 | 2020-05-04 11:39:30 | 144 | 157 | |
| 32 | 2020-02-15 10:14:20 | 51 | 60 | | 73 | 2020-05-04 12:36:04 | 102 | 149 | |
| 33 | 2020-02-15 10:44:14 | 38 | 59 | 9 | 74 | 2020-05-04 13:06:24 | 118 | 116 | |
| 34 | 2020-02-15 11:04:53 | 62 | 62 | 12 | 75 | 2020-05-05 08:46:10 | 224 | 212 | |
| 35 | 2020-03-05 12:10:54 | 45 | 27 | 11 | 76 | 2020-05-05 11:39:30 | 100 | 81 | 95 |
| 36 | 2020-03-05 12:27:04 | 32 | 27 | 16 | 77 | 2020-05-05 12:12:09 | 50 | 71 | 95 |
| 37 | 2020-03-22 08:54:08 | 82 | 77 | 108 | 78 | 2020-05-05 12:50:27 | 37 | 91 | |
| 38 | 2020-03-22 13:29:02 | 61 | 68 | 27 | 79 | 2020-05-06 08:08:55 | 62 | 153 | |
| 39 | 2020-03-22 13:54:00 | 47 | 64 | 30 | 80 | 2020-05-06 11:42:15 | 74 | 94 | |
| 40 | 2020-03-22 14:10:30 | 53 | 76 | 29 | 81 | 2020-05-06 12:09:51 | 78 | 111 | |
| 41 | 2020-03-22 14:17:36 | 72 | 77 | 31 | | | | | |