# Peer review of "Evaluation of methods to determine the surface mixing layer height of the atmospheric boundary layer in the central Arctic during polar night in cloudless and cloudy conditions"

_EGUsphere, 2023_

## Referee Comment (RC1)

**General Comments**

- Section 2.2: It's not entirely clear what you plan to do here. After reading through the entire section it seems you are describing the different methods available to calculate the SML height, and I think it's great to discuss the advantages and disadvantages of each method. But when introducing the section, I think it would help the reader to more explicitly state that you are going to use a variety of calculations and that you consider one of them to be a superior method given the instrument availability. Moreover, I think emphasizing the use of in-situ measurements to derive a critical bulk Richardson number for use with radiosonde measurements is prudent.

- Section 3.3: Can you provide any speculation as to why the critical bulk Richardson number estimates are so much smaller than the theoretical value?

- Section 3.5: I found the use of Fig. 11 alone to be rather confusing in the context of this discussion. The intent of the discussion appears to be describing the applicability of MO theory for reasonably locating the SML height. While I think it's interesting to point out where MO theory fails and why, this could be better captured by tweaking Fig. 11 to (1) show bounded regions, for example via shading, that are discussed in the text and (2) by perhaps using a twin axis on the top of each panel that corresponds to profile number. You often discuss the profile number, but one has to refer all the way back to Fig. 2 to know the profile numbers to which you are referring. In general, could your conclusions from Section 3.5 be summarized in a better way, for example something like repeating Fig. 8 (which made the comparison very clear)?

- The Summary and Conclusion section leaves a lot to be desired. I think you could elaborate quite a bit on what you intended to do with this study. It seems you are comparing various methods of identifying SML height using a turbulence-based method as ground truth. This is a unique aspect of this study that should be emphasized more. For example, when you discuss "confusion in the literature" on Line 344, this is the first time this supposed confusion is mentioned. I think you should be more explicit in stating that for this study and future MOSAiC studies, you determined that a critical bulk Richardson number of 0.12 was found and can be reliably used going forward (perhaps even give this exact value in the abstract). This was emphasized appropriately in the abstract but not enough in the final section, in my opinion. Moreover, the last paragraph is very short, but it seems like you are starting to speculate about remaining science questions that need to be addressed and how improved datasets can be used to address those questions. I think this is worthy of more discussion here. Discussion of future needs for estimating the SML height would also be useful. If the in-situ turbulence-based method is the best, what do future campaigns need in order to make this a reality?

- In general, the separation of cloudless versus cloudy boundary layers lacks any significant conclusions. I think readers would appreciate some extrapolation of these results with some physical interpretation. Why is determining the SML height for these two environments important? Are there modeling implications? Does it mean anything regarding cloud formation processes? As it stands, you seem to have arbitrarily separated the two, but the reader is left wondering why you did this in the first place.

- I commend the investigators for being brief and concise, but there is considerable room for further explanation, particularly regarding your methods and conclusions. I would take the time and space to elaborate quite a bit about your approach, the importance of it, and what the important conclusions are.

**Specific Comments**
- Line 23: Perhaps be a little more explicit regarding the vertical exchange under stable stratification. Vertical exchange of what? Aerosols? Heat? Momentum?
- Line 31: "Sea ice surface" itself is not a condition; perhaps rephrase to state what condition the sea ice surface produces that makes it unique
- Line 48: Become even more important for what? Some of these statements are very vague and could use more specificity.
- Line 65: I think you need to introduce what exactly the Richardson number is (ratio of buoyancy to shear) rather than just saying what it is used for. This is a big part of your study, but a brief description of how to interpret the Richardson number would be useful.
- Lines 70-71: Please include a reference for Monin-Obukhov similarity theory, and perhaps a brief description of what exactly it is.
- Line 99: You haven't yet defined the turbulence probe here, though I assume it is the hot wire anemometer package introduced on line 90. Perhaps specifically call this the turbulence probe on line 90.
- Lines 105-106: I don't think you've described the irradiance measurements/instrumentation yet.
- Line 116: You cite Illingworth et al. (2007), but provide no mention of what remote sensing instrumentation (lidar? radar?) or product from Cloudnet is used. In addition, you say you use Cloudnet but in the following sentence say that it poses challenges for low cloud layers. Is the cumulative surface radiative flux the only complementary measurement used? Regarding this, further down on line 120, you again mention radiation measurements but give no information on the instrument. I think some more information needs to be provided here.
- Line 142: You have not defined what "r" is in the equation for Taylor's hypothesis. Please do so and also provide a reference for Taylor's hypothesis, with perhaps a brief mention of what the hypothesis includes and why it is valid.
- Line 176: You have not defined what the distinct layers of "$\delta z$" are in this context. Please do so.
- Figure 3 caption and panel c: In the last sentence when you say that the SML heights derived by in situ turbulence are indicated by triangles, are you referring to the two left-pointing triangles on the right of the plot? Perhaps describe this a bit more clearly.
- Line 214: The first sentence here seems out of place, as you start discussing the bulk Richardson number for the cloudless profile and then move immediately toward discussing the cloudy profile. Suggest moving the first sentence to the discussion in the previous paragraph and starting this paragraph with the discussion of only the cloud profile.
- Fig. 9: You have an inlaid panel, I assume to emphasize structure during a small part of the observing period, but this panel is not discussed anywhere in the text nor in the

caption of the figure. I suggest either removing the inlay or specifically addressing it in the text.

- Line 304: It's rather hard to eyeball the specific 29 December event to which you are referring here. In addition, you reference profile #s 13 to 16, but this doesn't really mean anything since the x-axis of Fig. 11 is date and not profile #. Perhaps use a shaded bounding region in Fig. 11 to indicate this period that you explicitly discuss in the text.
- Line 308: "…if not vertical profiles are available." Vertical profiles of what, exactly? Cloudy profiles that may otherwise be detected by remote sensing instrumentation?
- Line 309: Again, referring to this specific time period is kind of hard for the reader. Suggest using a shaded bounding region to indicate this period to make the text discussion easier.
- Paragraph on lines 341-343: It is not common nor grammatically correct to end a paragraph with a colon and start with a new paragraph. Generally, if you provide a colon like this and intend to list specific points, the colon is followed by bullet points or a numbered series of points. Please address accordingly.

**Technical Comments**
- Line 23: "extend" should be "extent"
- Line 43: there should not be a comma between "clouds" and "impact"
- Line 69: "turbulence based" should be "turbulence-based"
- Line 70: Not sure what you mean by "evaluate the bulk Richardson number approach"; is the turbulence-based SML height used to evaluate the bulk Richardson number approach?
- Line 70: "surface based" should be "surface-based"
- Line 73: comma splice–use conjunction after comma
- Line 82: comma splice–use conjunction after comma
- Line 86: "The setup enables continuously vertical profiling" should be "The setup enables continuous vertical profiling"
- Line 94: I'm not sure what is meant by "an one component"; do you mean "a single component"?
- Line 100: "a few hundreds meters" should be "a few hundred meters" or "a few hundreds of meters"
- Lines 104-105: "taken at a meteorological tower in 2 m…height" should be something like "taken at a meteorological tower at 2m…heights"
- Line 128. I don't think there needs to be a comma after "then"
- Line 135: "...as turbulence measure" should be "...as the turbulence measure"
- Line 141: Do you mean angle brackets?
- Line 161: Comma splice; please include a conjunction after the comma
- Line 179: "…majority of the SML heights lies…" should be "…majority of the SML heights lie…"
- Line 180: "multi" should be "multiple"
- Line 202: Probably not a need for the word "two" before "cloudless"
- Lines 258-259: Here and in many other places in the manuscript, you are using "respectively" incorrectly, which requires one to list two features and two names attached

to those features. For example, the correct use here would be something like the following: "Figure 7b shows the distributions of Ribc for cloudless and cloudy conditions in orange and blue lines, respectively."

- Line 266: No need for a comma in this sentence.
- Line 300: Comma splice. Use a conjunction.

---

## Referee Comment (RC2)

Review of Manuscript egusphere-2023-629
"Determining the surface mixing layer height of the Arctic ABL during polar night in cloudless and cloudy conditions"
by E. F. Akansu et al.

**General comments**

This study examines the characteristics of the Arctic atmospheric boundary layer (ABL), more specifically the surface mixed layer (SML), during the Multidisciplinary drifting Observatory for the Study of Arctic Climate (MOSAiC) campaign, using measurements from a tethered balloon platform including in-situ turbulence measurements. The measured profiles of the turbulence dissipation rate are used to diagnose the depth of the SML. This diagnosis serves as a reference for the evaluation of the bulk Richardson method and the surface flux-based method. I think the most significant result of the study is the determination of a critical bulk Richardson number for the diagnosis of the height of the SML. Furthermore, two typical states of the Arctic ABL were observed and characterized: cloudless situations with a stable and shallow ABL, and cloudy conditions with a mixed ABL. The paper is nicely organized and mostly well written. The presentation is mostly clear. While I think the paper may be acceptable with only minor revisions following the comments below, I think it could be significantly strengthened following the suggestions of the other reviewer.

**Specific comments**

1. L23-L24: "... plays an important role as stable stratification hampers the vertical exchange and leads to a near surface warming contribution to Arctic amplification...". This is not clear to me. Wouldn't more stable conditions and reduce vertical exchange lead to surface cooling?
2. L172-L173: "For surface temperature, we use observations at the 2 m height". Would this not lead to a significant underestimation of the true surface inversion strength? Please comment on the differences.
3. L205: "with a less well mixed, neutrally stratified layer below the inversion" What do you mean? To me the sub-cloud layer looks like a rather well-mixed turbulent layer.
4. L208: "a slightly stably stratified layer". Seems still rather stable to me. Do you mean a "less stably stratified layer"?
5. L209-L210: "Near the surface, the wind speed increases with height, peaking at about 50 m again and continuing almost constantly until the maximum height of the profile." I find this sentence unclear, please reformulate.
6. L261: "the differences for the two mean case depending..." What do you mean? Please clarify.
7. Fig 3: The near surface conditions (with strong gradients) are difficult to see. You might want to increase the size of the panels.

**Technical comments**

8. L23: vertical extend --> vertical extent
9. L203: alternates between two cloudless and cloudy --> alternatives between cloudless and cloudy
10. L214: almost linearly increases --> increases almost linearly
11. L214: The new paragraph should start with "The example for vertical stratification under cloudy conditions..."
12. L221: decrease almost abruptly --> decrease quite abruptly
13. L231: As it is often --> As is often
14. L232: already begins clearly inside --> already begins inside
15. L237: near surface --> near the surface
16. L238: low values --> lower values
17. L241: turbulence continuously reaches --> turbulence reaches
18. L280: SML height smaller than 150 m --> SML height less than 150 m
19. L298: at the low humidities --> at low humidity values
20. Fig. 11: at surface --> at the surface

---

## Author Comment (AC1)

**Response to Anonymous Referee #1**

We appreciate the time and effort that you have dedicated to provide valuable feedback and helpful comments, which significantly improved our manuscript.

During the review process, we noticed an inaccuracy in the calibration of the hot-wire sensors that warranted a recalibration of the data. The changes are minimal and do not affect the conclusions of this manuscript, but all analysis were performed again with the re-calibrated data. The new data set is currently being in the publication process of PANGAEA and accepted as a data descriptor publication in Nature Scientific Data.

Here is the point-by-point response to your comments. We included the original comment in *blue and italics* with the lines referring to the first manuscript. Our response follows in black, and the line numbers refer to the revised version.

*This study evaluated various observing methods used to identify the surface mixed layer (SML) height during the Multidisciplinary drifting Observatory for the Study of Arctic Climate (MOSAiC) campaign, using in-situ turbulence measurements from a tethered balloon platform as ground truth for the SML height. The study finds a critical bulk Richardson number (~0.12) that can be extrapolated for use with radiosonde observations. This appears to be one of the main conclusions of the study and is worthy of documentation based on careful analysis by the investigators. However, I found the scientific exploration of cloudless versus cloudy boundary layers to be lacking and feel that the basis for using other methods (particularly Monin-Obukhov similarity theory) was not properly motivated. What can the community use from these conclusions going forward? I think emphasizing the derivation of a critical bulk Richardson number is indeed significant and important. However, I found little value in defining the differences in the SML height for cloudless versus cloudy boundary layers without any type of physical interpretation. Moreover, the manuscript was a little difficult to follow in terms of structure and most importantly in terms of grammatical fluency. Some of the figures (particularly Fig. 11) also seemed to display results/statistics that weren't entirely relevant and were hard to follow. While I think the scientific merit of the study is there and I commend the investigators for this, there needs to be some significant restructuring and improvements for it be acceptable for publication. I am therefore recommending reconsideration following major revisions.*

In response to your general comment, we have revised our introduction and elaborated on the motivation to distinguish between the two types of the Arctic atmospheric boundary layer (ABL): cloudless and cloudy conditions.
We agree also that using the Monin-Obukhov similarity approach was not explicitly explained and motivated, and we have now extended the description (see Sect. 2.2.3) to include this. We consider MO as a useful complement to estimate the surface mixing layer (SML) height for the time between airborne measurements. However, this is only possible if the boundary conditions for this method are met.
We agree that parts of the manuscript were difficult to read and hopefully have now improved the reading flow. Furthermore, we have followed your advice and improved Fig. 11, as this figure was quite difficult to interpret.

We hope that we have sufficiently addressed your minor and major concerns with our changes to the manuscript described below.

**General comments**

– **Section 2.2:** *It's not entirely clear what you plan to do here. After reading through the entire section it seems you are describing the different methods available to calculate the SML height, and I think it's great to discuss the advantages and disadvantages of each method. But when introducing the section, I think it would help the reader to more explicitly state that you are going to use a variety of calculations and that you consider one of them to be a superior method given the instrument availability. Moreover, I think emphasizing the use of in-situ measurements to derive a critical bulk Richardson number for use with radiosonde measurements is prudent.*

**Response:** Thank you for this valuable comment. We have followed your suggestion and summarized our approaches at the beginning of Section 2.2:

"In this chapter, we present several methods for determining SML heights based on different available data sets. First, we discuss a method to derive SML heights that benefits directly from local turbulence measurements and will serve as a reference for other approaches in the following (Sect. 2.2.1). A second method is based on the mean temperature and wind speed profiles in the context of the bulk Richardson number criterion (Sect. 2.2.2). Finally, we compare the previous results with SML heights based on the application of the Monin-Obukhov similarity theory and thus on the determination of surface fluxes (Sect. 2.2.3)." Lines 149-153

We have also expanded the introduction and already included a brief description of the different approaches there.

"By observing turbulence by in situ measurements, the SML height can be derived directly. One can either define the SML height as the height where $\varepsilon$ drops significantly with height, or when the flow is considered non-turbulent based on a "turbulence threshold" (Shupe et al., 2013; Brooks et al., 2017). [...] We use the turbulence-based SML heights as reference to derive a critical bulk Richardson number for the winter and spring. The in situ turbulence perspective not only allows to derive a critical value but also to evaluate the bulk Richardson number approach." Lines 71-83

– **Section 3.3:** *Can you provide any speculation as to why the critical bulk Richardson number estimates are so much smaller than the theoretical value?*

**Response:** For the bulk Richardson number method, the mean temperature and wind gradient of a layer is approximated by the difference from the surface to the layer height. It is assumed that the wind speed at the surface tends to zero, but the surface skin temperature is measured individually, and there are already different approaches in the application. By definition, the skin temperature should be used. However, this is rarely measured and the temperature at 2m height is used instead. This value, especially for radiosondes, is still subject to some measurement errors and, in particular, does not correspond to the skin temperature. The determination of the critical Richardson number is therefore directly related to the method used and is not universal. To show the influence of the method for calculating $Ri_b$, we have added the analysis of $Ri_b$ with the skin temperature as a surface reference. Although this does not explain all the differences between our critical value and the "theoretical value", it does show how much the critical Richardson number depends on the type of calculation and the surface reference used.

In Sect. 2.2.2, we mentioned the problem of the surface reference temperature by adding the following paragraph:

"According to the basic concept of the surface bulk Richardson number approach, the lower reference level is the surface where the mean horizontal wind velocity equals zero U (z = $z_0$) and $\theta_0$ is the skin temperature. Since other studies are often based on radiosonde observations only, the temperature at 2 m is then typically used for $\theta_0$, even though this value may differ from the skin temperature, especially for stably stratified conditions with strong surface temperature gradients. To be consistent with other studies, we first use temperature observations at 2 m from the *Met City* tower for $\theta_0$. Then, for comparison, the same analysis is performed using the skin temperature measured during MOSAiC." Lines 195-200

In Sect. 3.3, we have included the following paragraph to compare the influence of the different surface temperatures on the critical bulk Ri number:

"Applying the same analysis with the skin temperature as $\theta_0$, we derive $Ri_{bc}$ = 0.16 (Fig. 6). This difference shows clearly how strongly the derived value for $Ri_{bc}$ depends on the selected surface reference. Furthermore, we can assume that the temperature measured at a height of 2 m on a mast is also significantly more accurate than a corresponding measurement with a radiosonde, which has comparatively large inaccuracies in the lower ranges." Lines 283-286

– **Section 3.5:** *I found the use of Fig. 11 alone to be rather confusing in the context of this discussion. The intent of the discussion appears to be describing the applicability of MO theory for reasonably locating the SML height. While I think it's interesting to point out where MO theory fails and why, this could be better captured by tweaking Fig. 11 to (1) show bounded regions, for example via shading, that are discussed in the text and (2) by perhaps using a twin axis on the top of each panel that corresponds to profile number. You often discuss the profile number, but one has to refer all the way back to Fig. 2 to know the profile numbers to which you are referring. In general, could your conclusions from Section 3.5 be summarized in a better way, for example something like repeating Fig. 8 (which made the comparison very clear)?*

**Response:** We fully agree that Fig. 11 was confusing and have edited the figure to improve clarity according to your suggestions. We no longer show the temporal progress but have decided to show the profiles stacked as in Fig. 2. The x-axis now shows the profile numbers. The periods discussed in the text are highlighted with bounding and numbered in Roman. In addition, we have added a twin axis on top of the figure to give the month of the year. The shaded area in the plot shows the corresponding daylight condition as in the first version. To further improve the readability of the figure, we have decided to remove the Monin-Obukhov length *L* in a separate panel. In the text, we describe that *L* is used to derive $h_{MO}$ and that no $h_{MO}$ can be derived if *L* is negative.
While we agree that a scatter plot can be very helpful for interpretation, this is unfortunately not straightforward. The MO method complements the balloon perspective under certain conditions and fails when there is a second source of turbulence aloft (as in the case of clouds). These constraints complicate the interpretation of a scatter plot and, therefore, it is not possible to draw a general conclusion about whether the MO method over- or underestimates the SML heights. Thus, we aim to show that the MO method can bridge gaps between balloon measurements but is limited with respect to clouds and the occurrence of LLJs. We have added an explanation in Sect. 2.2.3.

"With the Monin-Obukhov scaling method, referred to as the MO method hereafter, the SML height can be derived continuously and complements balloon-borne SML height estimates. However, Eq. (5) is only valid for stable stratification ($B_s < 0$ and $L > 0$) and the method fails if further sources of turbulence are present at higher levels (such as clouds or Low-Level Jets (LLJ)). Again, the in situ turbulence method is helpful to assess the applicability of the MO method (Sect. 3.5)." Lines 225-229

Thank you for your valuable feedback. We hope that our changes have addressed your concerns.

— **Summary and conclusion:** *The Summary and Conclusion section leaves a lot to be desired. I think you could elaborate quite a bit on what you intended to do with this study. It seems you are comparing various methods of identifying SML height using a turbulence-based method as ground truth. This is a unique aspect of this study that should be emphasized more. For example, when you discuss "confusion in the literature" on Line 344, this is the first time this supposed confusion is mentioned. I think you should be more explicit in stating that for this study and future MOSAiC studies, you determined that a critical bulk Richardson number of 0.12 was found and can be reliably used going forward (perhaps even give this exact value in the abstract). This was emphasized appropriately in the abstract but not enough in the final section, in my opinion. Moreover, the last paragraph is very short, but it seems like you are starting to speculate about remaining science questions that need to be addressed and how improved datasets can be used to address those questions. I think this is worthy of more discussion here. Discussion of future needs for estimating the SML height would also be useful. If the in-situ turbulence-based method is the best, what do future campaigns need in order to make this a reality?*

**Response:** We strongly agree with your comment and have completely rewritten the "Summary and conclusion" section. Based on your comments, we have extended the discussion of the intention of our study and further developed its unique aspect with in situ turbulence-based SML height estimates. We followed your suggestion to include recommendations for further studies by discussing the advantages and disadvantages of the different methods and their limitations.

We finish the section now with our main conclusion:

"Finally we come to the conclusion that turbulence profile measurements are the most reliable method for SML height determination but at least profile measurements with radiosondes are necessary to either determine the SML height using the Richardson approach or to check the boundary conditions in combination with MO theory." Lines 422ff

— **General:** *In general, the separation of cloudless versus cloudy boundary layers lacks any significant conclusions. I think readers would appreciate some extrapolation of these results with some physical interpretation. Why is determining the SML height for these two environments important? Are there modeling implications? Does it mean anything regarding cloud formation processes? As it stands, you seem to have arbitrarily separated the two, but the reader is left wondering why you did this in the first place.*

**Response:** Thanks for your comment. We have addressed the separation between cloud and cloudy conditions in the Introduction and refer to the study of Solomon et al. (2023).

"We have distinguished between those two states of the atmosphere, as many models have difficulties to reproduce the bimodal distribution of the terrestrial radiation (Solomon et al., 2023)." Lines 39-40

As the height of the surface mixing layer is a parameter directly influenced by the presence of clouds, this separation was made to see whether a critical value to estimate the SML height using the bulk Richardson number approach needs to consider cloud conditions. We agree that this was not well explained in our first version of the manuscript and hope the reader is now adequately guided.

— **General:** *I commend the investigators for being brief and concise, but there is considerable room for further explanation, particularly regarding your methods and conclusions. I would take the time and space to elaborate quite a bit about your approach, the importance of it, and what the important conclusions are.*
**Response:** We agree and refer to our comment about the summary and conclusion which has been completely revised.

**Specific comments**

— **Line 23:** *Perhaps be a little more explicit regarding the vertical exchange under stable stratification. Vertical exchange of what? Aerosols? Heat? Momentum?*
**Response:** We have complemented this sentence to "the vertical exchange of energy" as this is the main driver of amplified warming in Arctic winter. Please see Lines 25-26.

— **Line 31:** *"Sea ice surface" itself is not a condition; perhaps rephrase to state what condition the sea ice surface produces that makes it unique*
**Response:** Thanks for your remark. We have changed this sentence:

"The Arctic ABL is formed under unique conditions, such as strong cooling of the sea ice surface and the lack of solar radiation during winter, which favors the evolution of stable atmospheric layering (Persson et al., 2002; Tjernström and Graversen, 2009; Morrison et al., 2012; Brooks et al., 2017)."
Lines 34-36

— **Line 48:** *Become even more important for what? Some of these statements are very vague and could use more specificity.*
**Response:** In the Arctic, long-lasting low-level clouds are especially important for the surface radiative energy budget as they alter the radiative fluxes at the surface. To name this directly, we have changed this sentence.
"While cloudless conditions are rather scarce in the Arctic (Intrieri et al., 2002b), frequently occurring low-level mixed-phase clouds are of major importance for the surface radiative energy budget." Lines 53-55

— **Line 65:** *I think you need to introduce what exactly the Richardson number is (ratio of buoyancy to shear) rather than just saying what it is used for. This is a big part of your study, but a brief description of how to interpret the Richardson number would be useful.*
**Response:** We have now extended the introduction to the bulk Richardson number and its use to derive the SML height based on a critical value. We hope that this makes the interpretation of the bulk Richardson number clear.
"Another approach to define the SML height applies the bulk Richardson number $Ri_b$ (Andreas et al., 2000; Zilitinkevich and Baklanov, 2002; Dai et al., 2014; Zhang et al., 2014; Jozef et al., 2022; Peng et

al., 2023). $Ri_b$ is derived from the ratio between shear and buoyancy and is a measure of whether turbulence tends to increase or decrease. The SML height is defined by the height where turbulence can not be sustained because the bulk Richardson number exceeds a critical value of $Ri_b$ (Zilitinkevich and Baklanov, 2002; Andreas et al., 2000). This critical value, however, is under discussion and varies, for example, among sites (Vickers and Mahrt, 2004)." Lines 75-80

— **Lines 70-71:** *Please include a reference for Monin-Obukhov similarity theory, and perhaps a brief description of what exactly it is.*
**Response:** We have added a reference to the Monin-Obukhov similarity theory:

"Further, continuous, surface-based energy flux measurements can be used to estimate the SML height using Monin-Obukhov similarity theory (Zilitinkevich, 1972; Vickers and Mahrt, 2004). The Monin-Obukhov similarity theory describes the near-surface turbulent exchange processes based on surface measurements. This approach can complement the balloon-borne SML height estimates between balloon launches." Lines 83-87

— **Line 99:** *You haven't yet defined the turbulence probe here, though I assume it is the hot wire anemometer package introduced on line 90. Perhaps specifically call this the turbulence probe on line 90.*
**Response:** You are right, thank you. We have revised this and introduced the term 'turbulence probe', which we use to refer to the hot-wire anemometer package.

"A hot-wire anemometer package specifically designed for turbulence observations (Egerer et al., 2019), hereafter referred to as 'turbulence probe', was used to measure the data of this study." Lines 107-108

— **Lines 105-106:** *I don't think you've described the irradiance measurements/instrumentation yet.*
**Response:** We recognize that we missed a proper description of the measurements and the instrumentation and added this in Section 2.1.

"Continuous, near-surface observations of meteorological and turbulence parameters were performed at a location on the ice floe called *Met City* (Shupe et al., 2022). Measurements with an ultrasonic anemometer/thermometer were taken at a meteorological tower at 2 m, 6 m, and 10 m heights serving as surface reference for our balloon observations. Furthermore, the upward ($\uparrow$) and downward ($\downarrow$), broadband terrestrial irradiances $F$ (in units of W m$^{-2}$) were collected at the Atmospheric Surface Flux Station with a Precision Infrared Radiometer (Shupe et al., 2022). The net irradiances $F_{net} = F^{\downarrow} - F^{\uparrow}$ are taken as a proxy for the radiative energy budget at the surface. Furthermore, the measurements at *Met City* include the surface radiometric skin temperature." Lines 124-130

— **Line 116:** *You cite Illingworth et al. (2007), but provide no mention of what remote sensing instrumentation (lidar? radar?) or product from Cloudnet is used. In addition, you say you use Cloudnet but in the following sentence say that it poses challenges for low cloud layers. Is the cumulative surface radiative flux the only complementary measurement used? Regarding this, further down on line 120, you again mention radiation measurements but give no information on the instrument. I think some more information needs to be provided here.*
**Response:** We agree that the information was not sufficient. Low-level clouds, as occurring frequently in the Arctic, are a known challenge for remote sensing instrumentation and therefore special

algorithms are applied. However, as we are interested in the effect of clouds on the surface radiative energy budget, we use the surface radiative flux measurements as a main cloud proxy. Additionally, we compare this to 360° all-sky photographs. Eventually, we decided to not use remote sensing instrumentation to characterize the cloud situation and, therefore, removed the Cloudnet information from the paper. Regarding the radiation measurements, please see the comment above.

"The cloudy Arctic ABL can be very shallow, especially in winter, and hence poses challenges on remote sensing approaches to detect the very low cloud layers (Griesche et al., 2020). For the question of the influence of clouds on the ABL dynamics, however, the net irradiances are of main importance, which are directly influenced by the clouds. Therefore, the terrestrial net radiation $F_{net}$ measurements at Met City, where $F_{net}$ is the cumulative surface irradiance, are used as an independent "cloud indicator". Cloudless conditions prevail when the net radiation is below -25 W m$^{-2}$, while higher values are associated with clouds (Wendisch et al., 2023). To avoid ambiguous allocations, data with net radiation in the range between -28 W m$^{-2}$ and -22 W m$^{-2}$ are not considered. Additionally, the cloud condition was manually compared with 360° photographs and all sky total imager observations (as far as possible regarding daylight conditions)." Lines 140-147

— **Line 142:** *You have not defined what "r" is in the equation for Taylor's hypothesis. Please do so and also provide a reference for Taylor's hypothesis, with perhaps a brief mention of what the hypothesis includes and why it is valid.*
**Response:** We now explained "r" as the 'spatial increment' and added information about Taylor's hypothesis, which allows to transfer time to length scales.

"The averaging in Eq. (1) denoted by the angle brackets is performed over all $t$, so the structure function $S^{(2)}$ is a function of time lag $\tau$ which has been calculated by applying Taylor's frozen turbulence hypothesis (transferring time lag to spatial increment $r = U \cdot \tau$ ) with the mean flow velocity $U$ (see Stull (1988) for more details)." Lines 162-165

— **Line 176:** *You have not defined what the distinct layers of "δz" are in this context. Please do so.*
**Response:** We have added "δz = 30 m".

"Compared to other bulk $Ri$-definitions, where mean gradients are estimated for distinct layers of δz = 30 m (Jozef et al., 2022), for example, the $Ri_b$-approach applied here fails if multiple turbulent layers are present." Line 202f

— **Figure 3 caption and panel c:** *In the last sentence when you say that the SML heights derived by in situ turbulence are indicated by triangles, are you referring to the two left-pointing triangles on the right of the plot? Perhaps describe this a bit more clearly.*
**Response:** Yes, that was the intention, and we hope the explanation we have given is now sufficient.

"[...] The SML heights for cloudless (orange) and cloudy conditions (blue) derived by in situ turbulence records are indicated by left pointing triangles on the right in (c)."

— **Line 214:** *The first sentence here seems out of place, as you start discussing the bulk Richardson number for the cloudless profile and then move immediately toward discussing the cloudy profile. Suggest moving the first sentence to the discussion in the previous paragraph and starting this*

*paragraph with the discussion of only the cloud profile.*
**Response:** Thank you for your suggestion, we have implemented it directly. Lines 243f

— **Fig. 9:** *You have an inlaid panel, I assume to emphasize structure during a small part of the observing period, but this panel is not discussed anywhere in the text nor in the caption of the figure. I suggest either removing the inlay or specifically addressing it in the text.*
**Response:** That is true and we recognize that we did not provide any explanation of the panel in the text. Originally, we have shown the cloud case of Fig. 3 in the inlaid panel but now decided to remove the inlay and updated Fig. 9.

— **Line 304:** *It's rather hard to eyeball the specific 29 December event to which you are referring here. In addition, you reference profile #s 13 to 16, but this doesn't really mean anything since the x-axis of Fig. 11 is date and not profile #. Perhaps use a shaded bounding region in Fig. 11 to indicate this period that you explicitly discuss in the text.*
**Response:** We fully agree with you and have revised the Fig. 11 as discussed above.

— **Line 308:** *"…if not vertical profiles are available." Vertical profiles of what, exactly? Cloudy profiles that may otherwise be detected by remote sensing instrumentation?*
**Response:** Indeed, we have missed an explanation. Thank you for your remark, we have added information on the measured parameters.

"Therefore, with rapidly changing cloud cover, SML height determinations using MO theory are only partially successful. $F_{net}$ observations can at least indicate these possible problems if no vertical profiles of thermodynamic or turbulent parameters are available." Lines 346-348

— **Line 309:** *Again, referring to this specific time period is kind of hard for the reader. Suggest using a shaded bounding region to indicate this period to make the text discussion easier.*
**Response:** We agree and followed your suggestions as described above.

**Technical comments**

We highly appreciate all your technical comments and have incorporated all suggestions you made here. In the following, we refer to the Lines in the revised manuscript, where this
— **Line 23:** *"extend" should be "extent"*
**Response:** Line 25
— **Line 43:** *there should not be a comma between "clouds" and "impact"*
**Response:** Line 49
— **Line 69:** *"turbulence based" should be "turbulence-based"*
**Response:** Line 81
— **Line 70:** *Not sure what you mean by "evaluate the bulk Richardson number approach"; is the turbulence-based SML height used to evaluate the bulk Richardson number approach?*
**Response:** Yes, this is exactly what we wanted to say here. We have changed this sentence and hope that this becomes clear now. Please see Lines 81 f.

"We use the turbulence-based SML heights as reference to derive a critical bulk Richardson number for the winter and spring. The in situ turbulence perspective not only allows to derive a critical value but also to evaluate the bulk Richardson number approach."

- **Line 70:** *"surface based" should be "surface-based"*

  **Response:** Line 84

- **Line 73:** *comma splice–use conjunction after comma*

  **Response:** We have rephrased the paragraph and removed the comma splice. Lines 88-92

- **Line 82:** *comma splice–use conjunction after comma*

  **Response:** We added a conjunction after comma. Line 98

- **Line 86:** *"The setup enables continuously vertical profiling" should be "The setup enables continuous vertical profiling"*

  **Response:** Line 102

- **Line 94:** *I'm not sure what is meant by "an one component"; do you mean "a single component"?*

  **Response:** Yes, we have meant a single component but have removed this information to prevent misunderstanding. Line 113

- **Line 100:** *"a few hundreds meters" should be "a few hundred meters" or "a few hundreds of meters"*

  **Response:** Line 119

- **Lines 104-105:** *"taken at a meteorological tower in 2 m...height" should be something like "taken at a meteorological tower at 2m...heights"*

  **Response:** Line 125f

- **Line 128:** *I don't think there needs to be a comma after "then"*

  **Response:** We have rephrased this paragraph. Please see Lines 149-153.

- **Line 135:** *"...as turbulence measure" should be "...as the turbulence measure"*

  **Response:** We have rephrased this sentence:

  "For our application, we chose the energy dissipation rate ε to quantify turbulence […]" Lines 156f

- **Line 141:** *Do you mean angle brackets?*

  **Response:** Yes, we do, thank you. Line 162f

- **Line 161:** *Comma splice; please include a conjunction after the comma*

  **Response:** We have added a conjunction. Line 185

- **Line 179:** *"...majority of the SML heights lies..." should be "...majority of the SML heights lie..."*

  **Response:** Line 205f

- **Line 180:** *"multi" should be "multiple"*

  **Response:** Line 206

- **Line 202:** *Probably not a need for the word "two" before "cloudless"*

  **Response:** We have removed it. Line 233

- **Lines 258-259:** *Here and in many other places in the manuscript, you are using "respectively" incorrectly, which requires one to list two features and two names attached to those features. For example, the correct use here would be something like the following: "Figure 7b shows the distributions of Ribc for cloudless and cloudy conditions in orange and blue lines, respectively."*

  **Response:** Thanks for pointing this out, we have revised the manuscript and removed the wrongly used "respectively".

- **Line 266:** *No need for a comma in this sentence.*

  **Response:** Line 300

- **Line 300:** *Comma splice. Use a conjunction.*

  **Response:** We have rephrased the sentence. Line 335

---

## Author Comment (AC2)

**Response to Reviewer #2**

We thank your for the valuable comments and appreciate the time and effort you dedicated to your feedback. These comments contributed to a significant improvement of our manuscript.

During the review process, we noticed an inaccuracy in the calibration of the hot-wire sensors that warranted a recalibration of the data. The changes are minimal and do not affect the conclusions of this manuscript, but all analysis were performed again with the re-calibrated data. The new data set is currently being in the publication process of PANGAEA and accepted as a data descriptor publication in Nature Scientific Data.

Here is a point-by-point response to the reviewers' comments. We included the original comment in *blue and italics* with the lines referring to the first manuscript. Our response follows in black, and the line numbers refer to the revised version.

**General comments**

This study examines the characteristics of the Arctic atmospheric boundary layer (ABL), more specifically the surface mixed layer (SML), during the Multidisciplinary drifting Observatory for the Study of Arctic Climate (MOSAiC) campaign, using measurements from a tethered balloon platform including in-situ turbulence measurements. The measured profiles of the turbulence dissipation rate are used to diagnose the depth of the SML. This diagnosis serves as a reference for the evaluation of the bulk Richardson method and the surface flux-based method. I think the most significant result of the study is the determination of a critical bulk Richardson number for the diagnosis of the height of the SML. Furthermore, two typical states of the Arctic ABL were observed and characterized: cloudless situations with a stable and shallow ABL, and cloudy conditions with a mixed ABL. The paper is nicely organized and mostly well written. The presentation is mostly clear. While I think the paper may be acceptable with only minor revisions following the comments below, I think it could be significantly strengthened following the suggestions of the other reviewer.

We thank you for reviewing our paper and appreciate your feedback and comments that improved the quality of our study. We have revised the manuscript considering the comments of both Anonymous referees with the hope that we have addressed all minor and major concerns. Please find the responses to your comments below.

**Specific comments**

 L23-L24: "... plays an important role as stable stratification hampers the vertical exchange and leads to a near surface warming contribution to Arctic amplification...". This is not clear to me. Wouldn't more stable conditions and reduce vertical exchange lead to surface cooling?

**Response:** This is somewhat misleading, the lapse rate effect describes the additional warming due to the fact that additional heat added to the system (by global warming) is distributed in a smaller volume (due to shallow SMLs) and therefore contributes to surface warming. This mechanism is described in detail by the cited work of Bintanja et al., 2011.

**L172-L173: "For surface temperature, we use observations at the 2 m height". Would this not lead to a significant underestimation of the true surface inversion strength? Please comment on the differences.**

**Response:** This is a very good point and indeed there is a strengthening of the inversion when the radiative skin temperature is considered. During MOSAiC the air temperature at 2 m height and the skin temperature were measured. We have added a second bulk Richardson number calculated with the skin temperature as surface reference to account for the inversion down to the surface.

"According to the basic concept of the surface bulk Richardson number approach, the lower reference level is the surface where the mean horizontal wind velocity equals zero U ( $z = z_0$ ) and  $\theta_0$  is the skin temperature. Since other studies are often based on radiosonde observations only, the temperature at 2 m is then typically used for  $\theta_0$ , even though this value may differ from the skin temperature, especially for stably stratified conditions with strong surface temperature gradients. To be consistent with other studies, we first use temperature observations at 2 m from the *Met City* tower for  $\theta_0$ . Then, for comparison, the same analysis is performed using the skin temperature measured during MOSAiC." Lines 196-201

We derived a critical value for both bulk Richardson numbers. The critical values are 0.12 for the reference temperature at 2 m and 0.16 when we use the skin temperature. Either temperature can be used depending on the purpose of the analysis. As skin temperature is not always measured, we concentrated on the standard meteorological parameters that are most likely to be measured during field campaigns. This approach should ensure applicability to other campaigns.

"Applying the same analysis with the skin temperature as  $\theta_0$ , we derive  $Ri_{bc} = 0.16$  (Fig. 6). This difference shows clearly how strongly the derived value for  $Ri_{bc}$  depends on the selected surface reference. Furthermore, we can assume that the temperature measured at a height of 2 m on a mast is also significantly more accurate than a corresponding measurement with a radiosonde, which has comparatively large inaccuracies in the lower ranges." Lines 284-287

L205: "with a less well mixed, neutrally stratified layer below the inversion" What do you mean? To me the sub-cloud layer looks like a rather well-mixed turbulent layer..
 Response: We fully agree and have change the sentence accordingly.

"The first case represents a cloudless ABL with a pronounced surface inversion, and the second case describes a cloudy ABL and the resulting elevated inversion at the cloud top with a well-mixed layer below the inversion." Line 234-236

 L208: "a slightly stably stratified layer". Seems still rather stable to me. Do you mean a "less stably stratified layer"?

**Response:** Yes, thank you, we have edited the sentence.

"Cloudless conditions prevailed during a profile observed on 5 March 2020 with a strong surfacebased temperature inversion ( $\Delta \theta \approx 7$  K within about 40 m) up to 50 m followed by a less stably stratified layer above (Fig. 3a)." Lines 238-239

- **L209-L210:** "Near the surface, the wind speed increases with height, peaking at about 50 m again and continuing almost constantly until the maximum height of the profile." I find this sentence unclear,

please reformulate..

**Response:** We agree and rephrased this sentence.

"The wind velocity *U* increases with height from the surface up to a height of about 50 m, and then remains almost constant up to the maximum height of the profile (Fig. 3b)." Lines 239-241

- **L261:** "the differences for the two mean case depending..." What do you mean? Please clarify.

**Response:** We referred to the two cases of cloudy and cloudless conditions but we recognize that this was not clear in that sentence. We have rephrased it.

"While we have derived Ribc for cloudless and cloudy conditions, the differences for the two typical ABL types are negligible." Lines 293-294

- **Fig. 3:** The near surface conditions (with strong gradients) are difficult to see. You might want to increase the size of the panels.

**Response:** Yes, that was rather hard to see in the first figure. We have enlarged the size slightly. We hope that this improves readability.

**Technical comments**

Thank you for your feedback on the language. We have incorporated all the suggested changes and provide the Line numbers below.

- L23: vertical extend --> vertical extent
   Response: Line 25
- L203: alternates between two cloudless and cloudy --> alternatives between cloudless and cloudy Response: Line 233
- L214: almost linearly increases --> increases almost linearly Response: Line 244
- L214: The new paragraph should start with "The example for vertical stratification under cloudy conditions..."

**Response:** Yes, we agree and have changed this accordingly. Please see Lines 243f.

- L221: decrease almost abruptly --> decrease quite abruptly Response: Line 251
- L231: As it is often --> As is often
   Response: Line 261
- L232: already begins clearly inside --> already begins inside Response: Line 262
- L237: near surface --> near the surface
   Response: Line 267
- L238: low values --> lower values
- Response: Line 268
- L241: turbulence continuously reaches --> turbulence reaches Response: Line 271
- L280: SML height smaller than 150 m --> SML height less than 150 m Response: Line 314
- L298: at the low humidities --> at low humidity values
   Response: Line 333
- Fig. 11: at surface --> at the surface
   Response: We have changed this, please see caption of Fig. 11.

---

## Author Response (AR2)

**Response to Reviewer #1**

We thank you for the comprehensive revision of our manuscript and your valuable and constructive comments. These comments have improved the clarity of our manuscript.

Here is a point-by-point response with the original comment in blue and italics. Our responses follow in black with line numbers referring to the revised version.

**Evaluation of methods to determine the surface mixing layer height of the atmospheric boundary layer in the central Arctic during polar night in cloudless and cloudy conditions**

*This paper evaluates methods for determining surface mixing layer (SML) height in the central Arctic using tethered balloon, radiosonde, and tower-based observations from the MOSAiC campaign. The SML height detection methods involve the energy dissipation rate, bulk Richardson number, and Monin-Obukhov similarity theory. The analysis is conducted considering both cloudless (with a surface-based inversion) and cloudy (with an elevated inversion) conditions.*

*Overall, the paper is well-written and easy to follow, and provides value to the scientific community. However, there are many points throughout the paper where I suggest the authors provide more detailed discussions, so the results can be better interpreted in context of the complex Arctic boundary layer dynamics. While I provide many specific comments below, they are all relatively minor in nature, and after consideration of these points, I would recommend this paper for publication.*

**General Comments:**

*Throughout the paper, there is some inconsistency between some methods/descriptions, and actual application, as it related to the time of year for the study period. For example, the title specifies that the study is for polar night, and throughout the intro and methods, ABL processes, and related surface energy budget are discussed in the context of what is relevant/true for polar night. However, Fig. 1 shows that some of the observations occurred during polar day. At other parts of the paper, it is explained that the study is conducted for cases both during polar night as well as during the transition to spring. This all needs to be cleared up. If the study period is really polar night AND spring, processes relevant to spring (e.g., the presence of solar radiation) need to be discussed and considered. Or if you only are using the cases during polar night and the transition to spring before the sun comes up (and not the cases during polar day that are shown in Fig. 1) this needs to be better clarified.*

Thank you for your comment. In our study we include both, the polar night and the transition to spring. We agree that the title was misleading and we have changed it accordingly: "Evaluation of methods to determine the surface mixing layer height of the atmospheric boundary layer in the central Arctic during polar night and transition to polar day in cloudless and cloudy conditions"

Further, we have clarified that the two states discussed in this study occur in winter and early spring (see e.g. Line 37). We also added some more description, e.g., "This is most pronounced during the polar night and gradually weakens in the transition to early spring." (Lines 43f)

**Specific Comments:**

L16: *This first sentence is a bit weak. The authors mention "intertwined mechanisms and feedbacks" and "Arctic climate parameters" without examples, so it comes across as very vague. This sentence could be strengthened by adding more descriptive terms.*

We agree that this sentence was vague and we have changed it as follows: "Currently, the Arctic climate is changing rapidly driven by intertwined mechanisms and feedbacks, such as the lapse-rate feedback and surface-albedo feedback, leading to an increased near-surface air temperature and corresponding sea ice retreat (Serreze and Barry, 2011; Wendisch et al., 2023a)." (Lines 16f)

L23: *It is unclear what you mean by "The ABL is the atmospheric layer above the Earth's surface whose effects are perceptible on small time scales." Do you mean that the effects of the Earth's surface on the ABL are variable on small time scales? If so, say that more clearly. If you mean something else, please clarify.*

Yes, you are right, this formulation left room for interpretation. We have simplified the sentence to "The ABL is the atmospheric layer above the Earth's surface that is directly influenced by the surface (Stull, 1988; Garratt, 1997)." (Lines 23-24)

L34: *It might be useful to add some more examples of how the Arctic ABL is unique. For example, the lack of convection most of the time, and the absence of a residual layer because there is no diurnal cycle of the sun for most of the year.*

We have added your examples and thank you for your comment.

"The Arctic ABL is formed under unique conditions, such as the strong cooling of the sea ice surface due to the lack of solar radiation during winter, which favors the evolution of stable atmospheric layering. Furthermore, the ABL does not develop a residual layer due to the absence of a diurnal cycle for most of the year and even during the polar day convection typically plays a minor role (Persson et al., 2002; Tjernström and Graversen, 2009; Morrison et al., 2012; Brooks et al., 2017). " (Lines 33-36)

L41: *Perhaps remind the reader here that you are referring to the processes during polar night. Or another option would be somewhere in the Intro to state that the processes described henceforth are characteristic of polar night (while there may be different processes at play during polar day that are not described).*

As we include both, the polar night and also the transition to spring, we added the following:

"This is most pronounced during polar night and gradually weakens in the transition to early spring." (Lines 43-44)

L46: *Clarify that this is the case with low clouds. If there are very high clouds, there can be little to no effect on the ABL or height of the inversion.*

Thanks, we have changed "clouds" to "low-level clouds". (Line 47)

L77: *A more true statement would be that Rib is a measure of the likelihood of turbulence to exist, where Rib below the critical value indicates an atmosphere that is likely to become or remain turbulent and Rib above the critical value indicates that an already laminar layer will not become turbulent.*

Thank you, we fully agree that this was not completely correct. We have followed your advice and changed the sentence accordingly.

"$Ri_b$ is derived from the ratio between shear and buoyancy and is a measure of the likelihood of turbulence to exist. A $Ri_b$ below the critical value indicates an atmospheric layer that is likely to remain or become turbulent. Turbulence cannot be sustained and laminar layers will not become turbulent if the $Ri_b$ is above a critical value." (Lines 78ff)

Throughout: *Sometimes you use the whole phrase 'bulk Richardson number' and other times you use the abbreviation 'Rib'. Should be consistent.*

We agree that there was still some inconsistency and have added: "In the following, $Ri_b$ always refers to the surface bulk Richardson number." in Line 203f for clarification.

L97: *The statement "in different fields" is vague. Do you mean to say that the measurement included those of the atmosphere, ocean, sea ice, biogeochemistry, and ecosystem? If so, say that.*

Yes, this is exactly what we meant and now we have added the respective details following your suggestion.

"The MOSAiC expedition facilitated measurements onboard RV *Polarstern* and on the ice floe covering the atmosphere, the Arctic ocean, sea ice, ecosystem, and biogeochemistry throughout an entire seasonal cycle." (Lines 98-99)

Figure 1: *I assume the white background in panel c indicates the brief period when there was a full diurnal cycle including day and night? You should clarify this by adding that to the legend, or stating that in the figure caption.*

We have added the information in the caption of Figure 1: "The background shading indicates the respective daylight conditions during the observation period with the white background indicating the period with a diurnal cycle."

L124: *Specify the distance between Met City and the balloon operations. This is important, considering that the greater the distance, the more likely that the two instrument sites are sampling a different or evolved airmass.*

This is a very good point and important for interpretation. The distance between Met City and the operation site of the balloon was always about 300 m but changed over time (according to Shupe et al., 2022, Figure 4 and Table 1 therein).

"There were about 300 meters distance between *Met City* and the balloon operation site increasing over time due to ice flow dynamics." (Lines 130-131)

L128: *It would be good to note that the net irradiance you are referring to is the net longwave (or as you say, terrestrial) irradiance, which is a proxy for radiative energy budget at the surface only during polar night when there is no shortwave (solar) radiation. However, regardless of the presence or lack thereof of solar radiation, the net longwave irradiance can be used to differentiate between cloudless and cloudy conditions. Clarify all of this in the text.*

We have added "terrestrial" here (Line 134). The explanation for using the net terrestrial irradiances as cloud indicator is given in Lines 148-149. We have followed your advice and added following sentence: "$F_{net}$ can be used to distinguish between cloudless and cloudy conditions during both polar day and night."

L139: *Jozef et al. (2022) showed that over 3 hours, when comparing UAS to radiosonde observations, the ABL height did not change significantly at the 5% significance level. You could add this reference to support your choice to compare coinciding balloon and radiosonde profiles, despite that atmospheric structure can change over this 3 hour time span.*

We agree and added the reference, thank you.

"The time difference between both launches is at most around 3 hours, in which the SML height typically did not change significantly as shown by Jozef et al. (2022)." (Lines 141-142)

L140: *A cloudless ABL can be even shallower than a cloudy ABL. This should also be mentioned when discussing observational challenges.*

While we fully agree that the even shallower cloudless ABL is challenging, we would like to point out here the cloud remote sensing challenge that very low Arctic clouds pose. The low-level clouds are often below the lowest detection limit of the cloud remote sensing instruments. To ensure that the reader relates shallower ABLs with cloudless conditions, we have edited the sentence: "The Arctic ABL can be very shallow, especially in winter and during cloudless conditions. However, also the cloudy Arctic ABL is often shallow, and hence poses challenges on remote sensing approaches to detect the very low cloud layers (Griesche et al., 2020)." (Lines 144f)

L177: *Some more description should be provided about how you examined the thermodynamic profiles and settled on the threshold value.*

We agree and have added the following more precise description:

"The highest threshold almost always yields the lowest $h_\varepsilon$ values, and the two lowest thresholds are very close and result in nearly identical values of $h_\varepsilon$. To identify the appropriate threshold value, we have compared the SML heights derived by different thresholds with the potential temperature profiles. Additionally, we have examined whether the SML height coincides with a significant change in ε." (Lines 180 ff)

L185: *Please clarify that values below the critical value imply turbulence.*

We agree and modified accordingly: "This stability measure describes whether there is a tendency for turbulence to weaken or strengthen. $Ri_g$ smaller than the theoretical value of 0.25 refers to an turbulent atmospheric layer." (Line 191)

L200: *It should be noted in the text that there could be a temperature offset between the met tower and the balloon due to a variety of reasons, e.g., the airmass evolved as it was advected between the two sites, or the two sites are sampling airmasses that were differently impacted by upwind features such as leads. This is a source of uncertainty in the Rib method that is explained.*

This is probably true but we consider this as a minor effect because the distance between *Met City* and the balloon operation site is only about 300 m.

L246: *How do you know that the elevated inversion base indicates cloud top in this example? You should provide some evidence to confirm this speculation, for example, from the MOSAiC ceilometer measurements. While what you say is likely true, this should be confirmed if you are going to make the definitive statement. I make this point because many studies have concluded that there is often a shallow stable layer between the ABL and the cloud, decoupling the two (e.g., Brooks et al., 2017).*

You are fully right that we have not confirmed our statement here and we decided to delete it because the exact location of the cloud is not important in this context.

Figure 4: *Is the red line the average? Please specify this in the caption for with a legend.*

Yes, this is the average and we have changed the figure and added a legend.

L294: *So have you shown here that the critical Richardson number does not necessarily vary based on atmospheric conditions (cloudless vs. cloudy)? Maybe state this as a conclusion, and note whether this agrees with any previous work, or is a new finding.*

We noted this in our conclusion, but it is difficult to say whether this is a new finding because other studies have used different definitions for the bulk Richardson numbers for cloudless and cloudy conditions (see Peng et al., 2023, for example).

L307-308: *What are the implications of this statement? If you are going to mention this, you should explain the potential impact on the results.*

We agree, this sentence was incomplete and we have added our intention: "It should be noted that the reference height $h_\varepsilon$ was not determined at the same time as the height based on the radiosonde ascents, which may explain some of the observed differences." (Line 313)

L315: *Change to "... around half of the profiles contained an SML with height less than 150 m"*

We edited the sentence accordingly, thank you.

L317: *What conditions do the tethered balloons miss? Based on previous discussion, it seems the balloon would miss the stormy conditions, where wind speeds are too high for the balloon to fly. In these cases, the SML is much deeper, likely related to the high winds and also the presence of clouds. Explain this, and the implications for the results. For example, your cloudy conditions in this paper are all those with relatively low winds. What might you expect to see when you have clouds AND high winds? Do you think that would change the critical Rib number?*

We agree and modified as follows: "The tethered balloon operations primarily cover periods of lower wind velocities and shallower SML heights, and during storm events the SML can be much deeper. Whether $Ri_{bc}$ may change during these events remains open and can not be answered in this study." (Line 321-323)

L325: *While different cloud characteristics can influence the SML differently, that is not the only factor at play here. Another important factor for SML depth is wind speed, which can also be highly variable. You could suggests here that the variation in SML during cloudy conditions may be affected by variations in wind speeds.*

This is a very good point and we directly added this thought to the manuscript:

"Furthermore, the wind velocity can play a role in the variation of the SML heights, as the wind velocity can vary significantly during cloudy conditions." (Lines 332-333)

Figure 11 caption: *"Further, shading refers to similar to daylight condition." Something sounds weird there.*

Yes, that was error-prone. We have changed the text in the caption of Figure 11 to: "[...] Further, shading refers to daylight conditions similar to Fig. 1. [...]"

L355: *Do you have an explanation for this atmospheric vertical structure? Perhaps advection of a warmer airmass at higher altitudes contributed to the elevated inversion?*

We don't have an explanation but this stratification is frequently observed in the high-latitudes during winter. We added a reference and modified the sentence: "With the beginning of the twilight (15 February 2020, profiles 34 to 36, period III), the cloudless ABL was characterized by a surface inversion and an elevated inversion above, separated by a weakly stable to neutral stratified layer in between, as frequently observed in high-latitudes during winter Mauritsen (2007)." (Lines 361ff)

L377: *This sentence doesn't make sense. Is there a typo?*

Yes, we corrected it.

L382: *It should also be noted that for cloudy conditions, warming of the near-surface atmosphere, relative to in the absence of clouds, lessens the suppression of shear-driven turbulence by the static stability (because the longwave cooling is reduced).*

According to your comment, we have added following sentence: "In addition, surface cooling, and hence stability, is reduced in the presence of clouds, leading to less suppressed wind-shear-driven turbulence." (Lines 391-392)

L388: *Rather that saying the cause is irrelevant, perhaps say something like "the cause for turbulence generation is not of concern for the current study."*

Thank you, we adjusted the formulation. (Line 397)

L390: *This method also requires the collection of skin temperature or 2 m temperature, in addition to the radiosondes, correct? This could be seen as a drawback as well, and should at least be noted, perhaps with a suggestion of what to do if no such measurements are collected.*

This a good suggestion and we added it as follows: "However, the $Ri_b$ method requires skin temperature measurements or reliable temperature measurements at a height of 2 m."

L395: *I find it hard to believe that not a single decoupled cloud situation was observed, as I would expect such conditions to occur occasionally throughout winter and spring. But you also did not include actual cloud height observations in this study to be able to make the claim that you make. So a more correct statement might be "For example, we were not able to quantify coupled versus decoupled clouds." I wonder if cloud coupling/decoupling could also explain some of the variability in SML height under cloudy conditions. In addition to repeating this study during the summer, you might also suggest for future work to add cloud observations so that the coupling/decoupling state can better be quantified.*

You raised a very good point and we fully agree that, first, our statement was too strong, and second, that cloud observations at the balloon would definitely add value to the discussion. We have changed the formulation (Line 406) and have extended the text with the following suggestion: "A valuable supplement to understanding the influence of coupling/decoupling is in situ cloud observations on the balloon." (Line 410f)

L425: *The sentence would read better as "... but profile measurements with radiosondes can also be useful to either..."*

Thanks, we changed it accordingly. (Line 436)